



# Development of a global dataset of Wetland Area and Dynamics for Methane Modeling (WAD2M)

Zhen Zhang[1,*], Etienne Fluet-Chouinard[2], Katherine Jensen[3,4], Kyle McDonald[3,4,5], Gustaf Hugelius[6,7],
Thomas Gumbricht[6], Mark Carroll[8], Catherine Prigent[9], Annett Bartsch[10,11], Benjamin Poulter[12]

[1]Department of Geographical Sciences, University of Maryland, College Park, MD 20740, USA
[2]Department of Earth System Science, Stanford University, Stanford, CA 94305, USA
[3]Department of Earth and Atmospheric Sciences, City College of New York, City University of New
York, New York, NY 10031, USA
[4]Department of Earth and Environmental Science, The Graduate Center, City University of New York,
New York, NY 10031, USA
[5]Carbon Cycle and Ecosystems Group, Jet Propulsion Laboratory, California Institute of Technology.
4800 Oak Grove Drive, Pasadena. CA 91001, USA
[6]Department of Physical Geography, Stockholm University, 10691 Stockholm, Sweden
[7]Bolin Centre for Climate Research, Stockholm University, 10691 Stockholm, Sweden
[8]Computational and Information Science and Technology Office, NASA Goddard Space Flight Center,
Greenbelt, MD 20771, USA
[9]CNRS, Sorbonne Université, Observatoire de Paris, Université PSL, Lerma, Paris, France
[10]b.geos Industriestrasse 1, 2100 Korneuburg, Austria
[11]Austrian Polar Research Institute, UZA1, Althanstraβe 14, 1090 Wien, Austria
[12]Biospheric Science Laboratory, NASA Goddard Space Flight Center, Greenbelt, MD 20771, USA

*Correspondence to*: Zhen Zhang (yuisheng@gmail.com)

**Abstract.** Seasonal and interannual variations in global wetland area is a strong driver of fluctuations in global methane (CH$_4$) emissions. Current maps of global wetland extent vary with wetland definition, causing substantial disagreement and large uncertainty in estimates of wetland methane emissions. To reconcile these differences for large-scale wetland CH$_4$ modeling, we developed a global Wetland Area and Dynamics for Methane Modeling (WAD2M) dataset at ~25 km resolution at equator (0.25 arc-degree) at monthly time-step for 2000-2018. WAD2M combines a time series of surface inundation based on active and passive microwave remote sensing at coarse resolution (~25 km) with six static datasets that discriminate inland waters, agriculture, shoreline, and non-inundated wetlands. We exclude all permanent water bodies (e.g. lakes, ponds, rivers, and reservoirs), coastal wetlands (e.g., mangroves and sea grasses), and rice paddies to only represent spatiotemporal patterns of inundated and non-inundated vegetated wetlands. Globally, WAD2M estimates the long-term maximum wetland area at 13.0 million km$^2$ (Mkm$^2$), which can be separated into three categories: mean annual minimum of inundated and non-inundated wetlands at 3.5 Mkm$^2$, seasonally inundated wetlands at 4.0 Mkm$^2$ (mean annual maximum





minus mean annual minimum), and intermittently inundated wetlands at 5.5 Mkm$^2$ (long-term maximum minus mean annual maximum). WAD2M has good spatial agreements with independent wetland inventories for major wetland complexes, i.e., the Amazon Lowland Basin and West Siberian Lowlands, with high Cohen's kappa coefficient of 0.54 and 0.70 respectively among multiple wetlands products. By evaluating the temporal variation of WAD2M against modeled prognostic inundation (i.e., TOPMODEL) and satellite observations of inundation and soil moisture, we show that it adequately represents

interannual variation as well as the effect of El Niño-Southern Oscillation on global wetland extent. This wetland extent dataset will improve estimates of wetland CH$_4$ fluxes for global-scale land surface modeling. The dataset can be found at http://doi.org/10.5281/zenodo.3998454 (Zhang et al., 2020).

## 1 Introduction

Wetlands cover about 10% of global land area (Davidson et al., 2018) and play an important role in regulating global climate

via biogeochemical cycling of greenhouse gases (IPCC, 2013). Wetlands are highly productive ecosystems that store large amounts of soil carbon due to their waterlogged conditions inhibiting aerobic soil respiration. Flooded conditions alter the soil redox state for microbes to favor methanogenesis and thus wetlands are the largest natural source of methane (CH$_4$) to the atmosphere, contributing ~20-30% of the total annual global methane budget (Kirschke et al., 2013; Saunois et al., 2016, 2020). The spatial and temporal distribution of wetlands is one of the most important and yet uncertain factors determining

the time and location of CH$_4$ fluxes (Melton et al., 2013; Parker et al., 2018). Wetlands are at risk from human activities such as land clearing and drainage, and also at risk from climate change caused drying or less predictable precipitation events (Davidson et al., 2018).

Because wetland definitions vary between science, applications and policy objectives, a definition suitable for CH$_4$ modeling is needed for comparative reasons and to avoid double counting. Since the first global wetland map of Matthews and Fung

(Matthews and Fung, 1987), several additional global and regional wetland area datasets have been developed (Table A1). These datasets are characterized by differences in definition, data sources, methodologies and time period covered. For example, the Ramsar Convention on Wetlands focusing on waterfowl conservation defines wetlands as both vegetated and non-vegetated systems (i.e., rivers, lakes, ponds). However, the biogeochemistry and methane flux pathways from open water and vegetated wetlands differs substantially. Additionally, human-made water bodies (e.g. reservoirs, rice paddies,

agricultural wastewater ponds (i.e., aquaculture (Grinham et al., 2018)) are considered wetlands in the definition of the IPCC National Greenhouse Gas Inventory guidelines (Hiraishi et al., 2014). The biogeochemical processes in these kinds of intensely managed wetlands differ from those of natural wetlands, and generic modelling approaches are not applicable. Boreal taiga forests and tropical floodplains, which are considered CH$_4$-emitting areas given their seasonally inundated states and significant CH$_4$ transport pathway via tree stem (Barba et al., 2019; Pangala et al., 2017), are omitted from many

wetland mapping products due to the difficulty in detecting dense forest canopies that hide surface inundation.



Broadly defined, wetland datasets available to this day fall into one of four types: 1) static maps based on a compilation of regional inventories based on geomorphic features and aerial photography (Finlayson et al., 1999; Hugelius et al., 2013; Lehner and Döll, 2004; Matthews and Fung, 1987; Wulder et al., 2018); 2) remote sensing derived products (Aires et al., 2017; Carroll et al., 2009; DeVries et al., 2017; Feng et al., 2016; Jensen and McDonald, 2019; Papa et al., 2010; Pekel et al.,

2016; Poulter et al., 2017; Prigent et al., 2001, 2007; Schroeder et al., 2015; Yamazaki et al., 2015); 3) prognostic hydrological water-balance modeling using approaches like TOPMODEL (Kleinen et al., 2012; Ringeval et al., 2010; Stocker et al., 2014; Zhang et al., 2016); 4) hybrid approaches that combine satellite observations with statistical modeling (Fluet-Chouinard et al., 2015; Gumbricht et al., 2017; Tootchi et al., 2019). These approaches differ in their representation of wetlands, ranging from long-term features of the landscape to area inundated at a given time.

Characterizing the seasonal and interannual variation in wetland extent is critical to improving global-scale wetland $CH_4$ modeling. Contemporary evidence from remote sensing (Alsdorf et al., 2000, 2007; Hu et al., 2018; Lunt et al., 2019; Melack et al., 2004; Pandey et al., 2020; Prigent et al., 2007, 2012; Rodell et al., 2018) and field monitoring (Dunne and Aalto, 2013) suggest that global wetlands, especially tropical floodplains, have a significant seasonal cycle and interannual variability in spatial extent that depend on changes in water balance (i.e. precipitation, runoff, and evapotranspiration) and

local topography. Despite the critical importance of spatial and temporal changes in wetland area, there are large discrepancies among the estimates of global wetland extent (Aires et al., 2018; Melton et al., 2013; Pham-Duc et al., 2017; Wania et al., 2013) and only a limited number of available global products characterize temporal dynamics in wetland extent (Gallant, 2015; Huang et al., 2014; Prigent et al., 2007, 2020).

Remotely sensed observations show potential for capturing spatio-temporal wetland patterns. While bottom-up inventories

define wetlands based on a combination of soils, hydrology and vegetation, satellite-based observations of surface inundation (i.e. water above the soil) capture areas that are permanently or seasonally wet. Microwave sensor-based products (Jensen and McDonald, 2019; Papa et al., 2010; Prigent et al., 2020; Schroeder et al., 2015) can sense water below vegetated canopies and now provide a multi-decadal records, with weekly-to-monthly revisit times. Optical sensor-based products using visible or infrared bands (Amani et al., 2019; Feng et al., 2016; Jones, 2019; Pekel et al., 2016; Wulder et al., 2018;

Yamazaki et al., 2015) observe the open water dynamics but have limited capacity to detect surface water beneath vegetation canopy. L-band (~1 GHz) synthetic aperture radar (SAR) can detect flooding beneath most vegetation canopies and is more successful at mapping forested wetlands than higher frequency observations such as optical or microwave products. These products separate inland water types at a high spatial resolution, but typically provide limited temporal coverage.

Data fusion approaches that merge remote sensing observations from multiple sources of sensors at different spatial

resolutions presents a feasible way to properly capture the dynamics of wetland extent. Despite recent progress in wetland mapping, long-term wetland dynamic datasets specifically suited for global $CH_4$ studies (Poulter et al., 2017) is an area of active research. Further, recent work has shown significant differences between remote sensing wetland products (Pham-Duc et al., 2017). These discrepancies can be linked to methodological differences (including pre-processing), data sources, and definitions. This introduces large biases in the modeling of wetland $CH_4$ emissions (Bohn et al., 2015), that can be traced to



the following limitations: 1) higher-spatial resolution optical sensors can only detect open water in the absence of clouds and vegetation (while SAR measurements can penetrate cloud and dense canopies but have inconsistent temporal coverage at the required wavelength); 2) available coarse-spatial resolution microwave based products accurately represent surface water only under low vegetation canopy cover conditions; 3) the intrinsic limitations in remote sensing include the difficulty in detecting inundation under snow cover. In addition, several recent studies (Fluet-Chouinard et al., 2015; Hess et al., 2015;

Prigent et al., 2007; Reschke et al., 2012) suggests that the wetland mapping products at coarse resolution tend to overlook small inundated areas. Some of the difficulty in merging these products arises from ambiguity in definitions of inundated versus open water wetlands. Also, widely used descriptions of wetlands (shallow water with depth less than 2-2.5m (Cowardin et al., 1979; Tiner et al., 2015)) overlap with a vast array of  lakes and small ponds - especially in permafrost peatlands and thermokarst regions (West and Plug, 2008). The confusion between wetlands and waterbodies  risks double-

counting $CH_4$ emissions from high-latitudes (Thornton et al., 2016). All these issues lead to biases and uncertainties in developing a global dataset of wetland extent.

The objective of this study is to develop a global dynamic wetland dataset with a data fusion approach using consistent definitions for use in wetland methane emission studies. Given the many wetland types used in the literature, we chose an operational definition of wetlands as all natural vegetated forested and non-forested wetlands, excluding coastal wetlands,

cultivated wetlands such as irrigated rice paddies, and open water systems such as rivers, streams, lakes, ponds, and reservoirs. Estimates of the methane producing area are used in all bottom-up $CH_4$-flux methodologies: from upscaling fluxes measured by eddy covariance at ecosystem scale (Knox et al., 2019; Peltola et al., 2019; Treat et al., 2018) to process-based modeling at global scale (Bloom et al., 2010; Melton et al., 2013; Poulter et al., 2017).

The resulting dataset, named Wetland Area Dataset for Methane Modelling (WAD2M), is designed to fuse multiple datasets

including ground-based wetland inventories, remote sensing products of open waters and surface inundation dataset based on optical and active and passive microwave satellite observations. Within this framework, the Surface Water Microwave Products Series (SWAMPS) is used as the basis for providing the temporal dynamics at a monthly timestep and at a spatial resolution of 0.25° over a 19-year period (2000-2018). A set of wetland-related datasets at different spatial resolutions representing lakes, ponds, rivers and streams, rice paddies, and a coastal mask, are applied to filter out non-vegetated and

anthropogenic wetlands. Another set of static maps representing non-inundated wetlands, such as peatlands, are used to fill-in the gaps of SWAMPS. Uncertainties are derived by comparing WAD2M with available benchmark products at regional and global scales.



## 2 Methods

### 2.1 Overview of data processing and wetland definition

Our data fusion approach begins with a time series of global, monthly surface inundation provided by SWAMPS v3.2 (Jensen and McDonald, 2019). The SWAMPS data set is derived from a series of active and passive microwave remote sensing observations used to estimate total area of surface inundation including all natural and managed terrestrial (open-to closed canopy vegetation) and open-water bodies, including coastal, lakes, rivers, ponds. All ancillary datasets (inventoried wetlands, remotely-sensed inland waters, rice, ocean) were re-gridded to 0.25-degree resolution to match SWAMPS and

expressed as fractional areas. The following sections describe the data processing in the following steps (Figure 1): The SWAMPS dataset was used to represent the temporal variation in wetland dynamics. For the wetland regions that were not captured or well-represented in SWAMPS mainly due to closed-canopy conditions, independent datasets of static wetland distributions were fused with SWAMPS. The merger was carried out in five steps: 1) by calculating the long-term maximum annual surface inundation from SWAMPS ($fw_{max}$), 2) on a per-pixel basis comparing $fw_{max}$ with the independent datasets of

static wetland distributions (see Methods 2.2), 3) adjusting $fw_{max}$ to match the wetland maps for pixels where $fw_{max}$ is less than the static distribution, 4) imposing the SWAMPS seasonal cycle to the corrected $fw_{max}$ dataset, and 5) removing inland water bodies, coastal waters, and rice agriculture.

We added missing wetlands to SWAMPS by fusing it with best available maps and inventories of under-represented wetlands separately across three latitudinal bands. For northern wetland inventories, we used the Northern Circumpolar Soil

Carbon Dataset (NCSCD; (Hugelius et al., 2013) to map permafrost and non-permafrost peatlands (Histels and Histosols). Mineral soil wetlands were mapped from SAR-based map by including occurrences of wetlands in the circum-arctic (Widhalm et al., 2015) outside areas mapped as peatlands by the NCSCD. In the tropics, we used a 231-m resolution pan-tropical dataset based on geomorphic classification approach (Gumbricht et al., 2017). For temperate regions not covered by either the boreal and tropical datasets, we used the 1-km Global Lakes and Wetlands Dataset (GLWD) Level 3 after

removing Classes 1-3 lakes and rivers (Lehner and Döll, 2004). The global dataset of Monthly Irrigated and Rainfed Crop Areas (MIRCA2000) at 10-km resolution, was used to remove rice agriculture (Portmann et al., 2010). Lakes, ponds, rivers and other permanent inland water bodies were removed using the Landsat Global Surface Water dataset (Pekel et al., 2016). An ocean/coastline mask based on MOD44W Collection 6 (Carroll et al., 2009), a 250-m resolution annual product from the Moderate Resolution Imaging Spectroradiometer (MODIS) remote sensor, was used to remove ocean waters. The new

SWAMPS v3.2 (Jensen and McDonald, 2019), is an updated version over SWAMPS v2.0 (Schroeder et al., 2015) that was used  as input in the hybrid wetland product SWAMPS-GLWD (Poulter et al., 2017), the predecessor of WAD2M. The major differences between WAD2M and SWAMPS-GLWD are that 1) WAD2M uses an updated version SWAMPS v3.2 with improved algorithm and ancillary datasets; 2) WAD2M uses multiple static wetland maps as mergers in the processing (while SWAMPS-GWLD only considers GLWD in the processing); 3) The WAD2M includes removal of lakes, ponds,

rivers, streams and irrigated rice paddies, and 4) WAD2M uses a globally consistent ocean/land mask.



To characterize the temporal dynamics, three wetland statistics were computed: (1) Mean Annual minimum ($MA_{min}$); (2) Mean Annual maximum ($MA_{max}$); (3) Long-term Annual Maximum ($MA_{Lt}$). For each 0.25-degree grid cell, annual magnitude in wetland area can be calculated as difference between $MA_{max}$ and $MA_{min}$, while wetland areas that do not flood during the average year (i.e., intermittent wetlands) can be calculated as difference between $MA_{Lt}$ and $MA_{min}$.

## 2.2 Datasets

### 2.2.1 Wetland Dynamic Dataset

The Surface Water Microwave Product Series v3.2 (SWAMPS) is a long-term, daily time series of inundated area fraction dataset derived from microwave remote sensing. The SWAMPS dataset provides estimates of terrestrial surface water dynamics, including for wetlands, rivers, lakes, ponds, reservoirs, rice paddies, and episodically inundated areas. SWAMPS provides estimates of global inundated area fraction (fw) developed under the NASA Making Earth System Data Records for Use in Research Environments Program (MEASURES). SWAMPS *fw* estimates are derived from a combination of passive microwave brightness temperature and active microwave radar backscatter from a variety of satellite sensors supplemented with *a priori* knowledge of land cover based on a static MODIS land cover product (Schroeder et al., 2015). The sensors used in SWAMPS product include daily gridded DMSP Special Sensor Microwave Imager-Special Sensor Microwave Imager Sounders (SSMI-SSMIS) Pathfinder brightness temperature observations and active microwave backscatter from NASA SeaWinds-on-QuikSCAT Level 1B Sigma0 Product and Advanced Scatterometer Level 1B (ASCAT) product, with ancillary snow water equivalent, land cover map and NDVI from AVHRR and MODIS for delineating snow cover and arid and semiarid areas. SWAMPS v3.2 is an update of v2.0 and includes a new cloud and snow mask, a quality control flag, a new land and ocean mask, freeze-thaw detection, and improved sensor intercalibration. For the purpose of this study, the SWAMPS v3.2 dataset, covering the years 2000 to 2018, were merged into a single time series using samples flagged as 'Valid Observations'. For SWAMPS v3.2, the coastal zone was filtered out using a Landsat-based 90-m mask of permanent ocean waters defined by the G3WBM Global Water Body Map dataset (Yamazaki et al., 2015) – but later re-filtered using the MODIS MOD44W product. The SWAMPS v3.2 data were remapped to WGS84 using bilinear interpolation at 0.25-degree resolution with values aggregated from daily to monthly means.

### 2.2.2 Open water & land-ocean masks

The Global Surface Water (GSW) product is derived from 16-day Landsat thematic mapper imagery at 30-m spatial resolution and identifies the presence or absence of water bodies over the period 1984-2016 (Pekel et al., 2016). We used this dataset to represent permanent water bodies which we define as those covered by open water for more than 50% of the months during this time period. We used this as a permanent waterbody mask to avoid including temporary waterbodies that are considered wetlands in our working definition. This distribution of long-term maximum permanent water was re-gridded to 0.25-degree fractional area per grid cell and used for removing inland-water areas from SWAMPS v3.2. Because the





coastal regions were masked out in the processing of SWAMPS, we used the MODIS product MOD44WC6 (Carroll et al., 2009) to generate an ocean mask in the processing of GSW to avoid over-deducting. The coastline was buffered by 4 pixels (~1 km) into the water bodies. The buffered water was intersected with the ocean-labelled pixels from MOD44WA1 to

separate the ocean from inland water. The resulting ocean mask was then applied to remove coastal wetlands in GSW. The static long-term open water area excluding coastal regions in GSW is 4.5 Mkm$^2$, compared with the river and stream surface areas of 0.8 Mkm$^2$ (Allen and Pavelsky, 2018).

### 2.2.3 Static wetland distributions

We used static wetland maps to fill gaps left by wetland types that are under-represented or missed by the SWAMPS dataset.

However, most static maps do not have global coverage or tend to have lower accuracy compared to the regional products, leaving us to take a separate merging approach for each of three latitudinal bands.

Many arctic wetlands, including peatlands do not have surface inundation and thus are not captured by SWAMPS 3.2, but still emit methane. We use the Northern Circumpolar Soil Carbon Dataset (NCSCD) to map permafrost and non-permafrost peatlands based on the Histels and Histosols soil orders (Hugelius et al., 2013). The NCSCD dataset is a digital polygon-

based database compiled from harmonized regional soil classification maps in which data on soil order coverage have been linked to pedon data. In this study, the NCSCD wetland distribution is used as supplementary data for the latitudinal bands from 60°N-90°N. In this study we use a gridded version with a spatial resolution at 0.25 degrees. Permafrost and non-permafrost peatlands (Histels and Histosols, defined as >40 cm surface peat) are mapped in the NCSCD from harmonized regional and national soil maps (Hugelius et al., 2013). However, these maps do not include occurrences of mineral soil

tundra wetlands (with organic soil horizons of 0 to 40 cm) and the maps do not include smaller wetland complexes (Hugelius et al., 2020). To better include these types of wetlands, the NCSCD soil maps were combined with CircumArctic Wetlands based on Advanced Aperture Radar (CAWASAR) by Widhalm et al., (2015). The SAR data identifies both organic and mineral wetland soils. It is based on ENVISAT Advanced SAR data acquired in Global Monitoring mode (medium resolution) under frozen soil conditions, what represents surface roughness which can serve as proxy for wetness levels in

tundra. The wettest class was included as wetland. It corresponds to soils with >25 kg C m² in the top 100 cm (Bartsch et al., 2016). To avoid double counting of organic wetlands (peatlands) the datasets were overlayed so that any overlap between the datasets was removed, maintaining the NCSCD in the output data. The merged static map covers 2.3 Mkm$^2$ for the high latitudes (>60°N), including peatlands and mineral wetlands in the tundra biomes.

The distribution of tropical wetlands, including annually or seasonally water-logged area and tropical peatlands, are derived

from an expert-system mapping product (Gumbricht et al., 2017). We used the CIFOR wetland distribution for adjusting wetlands in the latitudinal bands from 60°S-40°N. This static map was generated by combining satellite images, and Topographic Convergence Indices (TCI) by the Center for International Forestry Research (CIFOR). The TCI indices are calculated based on Shuttle Radar Topography Mission (SRTM) Digital Elevation Model (DEM) at 250 m resolution with





precipitation climatology from WorldClim global data set (Hijmans et al., 2005). A simplified hydrological model was used
to estimate the local vertical water balance, runoff, and estimate flood volumes. The topographic and hydrologic data are
merged with MODIS (MCD43A4) images used for estimating the duration of wet and inundated soil conditions. The
estimated area of tropical peatlands and wetlands are ~1.7 Mkm$^2$ and ~4.7 Mkm$^2$ respectively. The estimated extent
of CIFOR for the Cuvette Centrale tropical African peatland in the Congo basin is 125,400 km$^2$, which is in agreement with
145,500 km$^2$ of a recent independent field investigation (Dargie et al., 2017).

The Global Lakes and Wetland Dataset (GLWD) (Lehner and Döll, 2004) is a global database of lakes, reservoirs, and
wetlands based on the aggregation of aerial surveys, surveyor maps and inventories at global and regional scales. While
GLWD was generated from data sources now decades old, for some regions, it still represents the most complete wetland
database available today. In this study, the GLWD wetland distribution is used to cover the temperate wetland only in the
latitudinal band 40°N-60°N, outside the range of NCSCD and CIFOR. We used the Level 3 product, a global raster map that
contains 12 classes of waterbodies and wetlands at the 30-second resolution. We excluded the classes representing lakes,
rivers and reservoirs (1-3) and estimated the area of fractional wetland classes (9-12) as the midpoint from the range of each
class. We then calculated the total fraction of wetland from all classes in 0.25-degree pixels. The estimated total wetland
extent in GLWD is 8.7 Mkm$^2$ for the globe and 2.7 Mkm$^2$ for the 40°N-60°N bands.

### 2.2.4 Irrigated rice distributions

The distribution of rice paddies is derived from the global data set of monthly irrigated and rainfed crop areas for the year ca.
2000 (MIRCA2000) (Portmann et al., 2010). The datasets used to develop MIRCA2000 are based on compiling census-
based land use datasets downscaled to grid-cell level and thus is generally consistent with subnational statistics collected by
national institutions and by the FAO (Food and Agriculture Organization of the United Nations). For this study, we extracted
the annual maximum area of irrigated rice paddies from its original resolution at 5 arc-minute and remapped to 0.25-degree
245 resolutions. We did not consider rainfed rice as we could not reliably separate lowland from upland cropping practices, with
only the latter seasonally contributing to surface inundation. The estimated rice paddies in MIRCA2000 (irrigated: 0.64
Mkm$^2$; rainfed: 1.13 Mkm$^2$) is largely consistent with census-based national and sub-national statistics from FAO (1.54
Mkm$^2$ for total area at ca. 2000) and slightly lower than a remote sensing estimate for irrigated (0.66 Mkm$^2$) (Salmon et al.,
2015),. We thus apply the monthly rice cover from 2000 across the entire 2000-2018 time-series. This assumption ignoring
250 year-on-year change in rice paddy area is reasonable given that its area increased by < 1.6% over 2000-2017 according to the
IRRI World rice statistics (http://ricestat.irri.org:8080/wrsv3/entrypoint.htm).



## 2.3 WAD2M evaluation

The WAD2M was evaluated against several, both static and dynamic, independent datasets of wetland area and surface
inundation (Table A1). We used a set of satellite-based terrestrial water dynamics to evaluate the trends in temporal pattern
of WAD2M, including (1) global soil moisture time series based on ESA Soil Moisture and Ocean Salinity mission (SMOS;
level-4; (Kerr et al., 2012); (2) a global inundation time series from the Global Inundation Extent from Multi-Satellite
(GIEMS version 2) (Prigent et al., 2020); (3) a global land water mass dynamics product from the Gravity Recovery and
Climate Experiment mission (GRACE; (Landerer and Swenson, 2012); and (4) a global inundation dynamics from a
prognostic run of a land surface model LPJ based on topography-based hydrological model (TOPMODEL) using CRU
meteorological forcings (Zhang et al., 2018). We also compare to a global static map from Tootchi et al., 2019 (regularly
flooded wetlands plus groundwater-driven wetlands based on topographic index; hereafter denoted as Tootchi2019) and
region static maps available over the West Siberian Lowlands (Terentieva et al., 2016) and Amazon Basin (Hess et al., 2015).
The similarity of WAD2M performance to these the independent validation data is evaluated using the Kappa index.

## 3 Results and Discussions

### 3.1 Effect of data processing on the results

Globally, WAD2M (MAmax) identifies 3.6 Mkm$^2$ more wetlands compared to SWAMPS v3.2 (Table 1). On a continental
scale, the wetland extent of SWAMPS v3.2 is in general agreement with inventories except for pronounced discrepancies for
Tropical wetlands (e.g. Amazon Lowland and tropical Africa), central Asia, and the Sahel regions. The lower area of tropical
wetland in the SWAMPS v3.2 is generally due to the influence of dense forest canopies. It should be noted that the
SWAMPS v3.2 detected higher wetland area in India than southeastern China, due to the inclusion of rice paddies in
SWAMPS v3.2 that are masked out in WAD2M. Fig. 2d shows the comparison of the latitudinal gradient between original
SWAMPS and WAD2M. Generally, WAD2M maintains the same latitudinal pattern as the original SWAMPS, where the
peak of wetland area occurs in latitudinal bands of 40☐N-50☐N for boreal wetlands and another peak around the equator for
tropical wetlands.

Table 1 quantifies the effect of the data processing steps on the continental and global estimates of wetland area. The total
area including all water bodies such as rice paddies, rivers, streams, lakes, ponds, and reservoirs after $fw_{max}$ correction are
17.0 Mkm$^2$ for MA$_{Lt}$. This number is close to the downscaled GIEMS-D15 (17.3 Mkm$^2$), also produced through data merger,
suggesting a good agreement between the two products. Applying the $fw_{max}$ correction leads to a ca. 20% increase for the
three states of inundation relative to the SWAMPS v3.2. As intended, the augmentation with inventories filled many missing
or underestimated wetland areas of the SWAMPS dataset, which include the Congo floodplain, Amazon Basin lowlands, the





Pantanal, Southeast Asia peatlands, and peatlands in high latitudes (i.e., Hudson Bay Lowlands and Western Siberian Lowlands). The highest increase of wetland areas between SWAMPS v2.0 and SWAMPS v3.2 occurs in Asia, followed by North America in the $fw_{max}$ correction step. However, when we subsequently removed open water and rice paddies in the last

step, the increase in wetland area for Asia, North America, and Europe are eliminated. As a result, only South America has a higher wetland area in WAD2M than in SWAMPS v3.2.

## 3.2 Spatial distributions

### 3.2.1 Global distributions

Our estimated total annual maximum area of global vegetated wetlands (excluding Greenland and Antarctica) is ~13.0 Mkm$^2$

(Fig. 3a). This estimate consists of 3.5 Mkm$^2$ of mean annual minimum (Fig. 3b), 4.0 Mkm$^2$ of seasonally inundated wetlands (MA$_{max}$ minus MA$_{min}$) (Fig. 3c), and 5.5 Mkm$^2$ of intermittently inundated wetlands (MA$_{Lt}$ minus MA$_{max}$) (Fig. 3d). Our estimated global total wetland area is slightly higher than GIEMS2 (Table 2) but is lower than a high-resolution version of GIEMS initial version GIEMS-D15, which reports a long-term maximum of 17.3 Mkm$^2$ (Fluet-Chouinard et al., 2015). Considering that WAD2M conservatively excludes rice paddies (0.59 Mkm$^2$), rivers, streams, and lakes and ponds (2.52

Mkm$^2$) while GIEMS-D15 include these water bodies, one possible conclusion is that WAD2M applies the upward mergers of CIFOR and NCSCD, which has lower wetland estimates than GLWD, causing a lower long-term maximum than GIEMS-D15. In addition, our estimated total area for intermittently inundated wetlands is close to the 5.2 Mkm$^2$ reported for similar wetlands by GIEMS-D15, suggesting a good agreement for temporary inundated areas between two independently developed products. Other recent studies (Hu et al., 2017; Tootchi et al., 2019), however, proposed a much higher global

wetland area of 27-29 Mkm$^2$, which are likely overestimations due to their approaches based on topographic wetness indexes that do not take into account the location of surface-water tables. This leads to an overestimation of the inundated area with shallow groundwater tables, and large inundated areas in e.g. Central Asia and South America that are not matched by other wetland maps.

Permanently inundated wetlands are located in well-documented wetland hotspots, including the Hudson Bay Lowland and

West Siberian Lowland, where a large extents of peat bogs and fens are not represented by SWAMPS v3.2. In tropical regions, key peatland areas along the Amazonian floodplain, in the Cuvette Centrale of the Congo (Dargie et al., 2017), and the tropical peatlands in Indonesian Papua are all captured by WAD2M. The subtropical and boreal regions are the main contributors to the seasonally inundated wetlands. For the subtropical regions, the seasonal wetlands are largely located in Southeastern Asia and the Sahel, where the variation of wetlands is mainly driven by the annual cycle of precipitation in

monsoon regions. For the high latitudes, seasonal wetlands are primarily in the transition region from temperate to boreal in across both North America and Eurasia. The high seasonality of inundation in these regions were also captured by the surface inundation retrievals of passive microwave observations from the Soil Moisture Active Passive mission (SMAP) (Du et al., 2018).





The latitudinal distribution of wetland area (Fig. 4) suggest that the northern hemisphere mid-to-high latitudes (> 45°N) have
the highest coverage of wetland area with 45±5% of the total area of wetlands, followed by the equatorial region (10°S-10°N). A large portion of the intermittent wetlands are found in the northern mid-high latitudes, in regions that also have large areas of seasonal wetlands. The overall latitudinal pattern in WAD2M is similar to that of other estimates except for the Tootchi2019, which has the highest wetland area along the latitude gradient. The exception is over the mid-latitudes (20°N-40°N) where the wetland area in GLWD are more extensive than that in WAD2M. The wetland areas in the arctic (>60°N)
in WAD2M have lower wetland extent than GLWD and NCSCD but higher than GIEMS2. The WAD2M shows a slightly higher wetland extent in the latitudinal band of 10°N-15°N compared to the other products, which we attribute to the higher intermittent wetlands in Southeastern Asia detected by SWAMPS (Fig. 3d).The latitudinal gradient of the wetland area in WAD2M is similar to the previous version SWAMPS-GLWD (Poulter et al., 2017), but with a reduced wetland area in the Arctic (> 50°N) and at mid-latitudes (15°N-45°N), a consequence of the masking out the inland-water areas from GSW.
Surface inundation products (GIEMS2 and SWAMPS) have limited observations in the high latitudes due to underestimates of wetland extent for unsaturated peatlands (Bohn et al., 2015), the presence of snow and ice, and are not reliable points of comparison in high latitudes.

### 3.2.2 Regional comparison

We validated WAD2M against available independent fine-resolution datasets for the two methane emitting hotspots,
Amazon Basin Lowlands (defined as the portion of the Amazon watershed below 500 m asl.) and West Siberian Lowlands. These two regions represent different wetland subtypes, vegetation compositions and local hydrology, making them complimentary for our validation.

The distribution of wetland area from WAD2M shows a similar spatial pattern for the Amazon Basin Lowlands compared to the map based on JERS-1 SAR (Hess et al. 2015), which was used by Pangala et al. (2017) to estimate methane emissions.
WAD2M have a good similarity (kappa=0.54) with the independent, L-band synthetic aperture radar (SAR) map, slightly lower than GIEMS2 (kappa=0.56; Fig. 6a) but higher than all other global products compared (range: 0.1-0.2). WAD2M adequately captures the permanently inundated wetlands along with the Amazon Basin river channel network as well as temporarily flooded wetlands during the wet season (Fig. 5a). However, considerable spatial disagreements of the wetland location and extent were found among available datasets when compared with Hess et al., 2015. The disagreements captured
in Fig. 6a are primarily related to the seasonally inundated Pantanal floodplains that have relatively flat terrains with dominant herbaceous/shrubland (height < 5 m) and the Pastaza-Marañon wetland basin of the Western Amazonia that have large areas of permanently inundated forested swamps with dense-canopy. The WAD2M estimated wetland fraction exhibits a reasonably good agreement with independent SAR-based wetland products from Hess et al., (2015) for the Pantanal floodplains and the Ucayali-Maranon wetlands basin. The WAD2M estimates for Pastaza-Marañon swamps are close to
CIFOR and Hess et al., (2015), and are considerably lower than Tootchi2019. Over the Pantanal floodplains, WAD2M shows moderate-density wetlands while the Tootchi2019 suggest a widespread high-density area over the same extent. The





CIFOR estimate is likely an underestimation given the limitations of its topographical hydrology approach at estimating inundation over flat terrain like the Pantanal.

The comparison of multiple wetland mapping products for the West Siberian Lowland (Fig. 5b) shows that WAD2M permanent wetlands capture the general spatial distribution represented by most other products. WAD2M shows a good agreement with the independent dataset from Terentieva et al., 2016 that combine field survey and satellite images, in particular for inundation peatlands (e.g. fens and bogs) from 55°N-65°N in the Taiga forests. The Kappa coefficient of the WAD2M wetland map with Terentieva et al., 2016 for the West Siberian Lowland is 0.70 (Fig. 6b), higher than the value of GIEMS2, GLWD, SWAMPS-GLWD, Tootchi et al., 2019 (Kappa coefficient of 0.18, 0.57, 0.54, and 0.43 respectively).

Wetlands in high latitudes above 65°N are more intermittent, caused by thawing of permafrost and thus related to interannual climate variations.

### 3.3 Temporal patterns

### 3.3.1 Seasonal cycle

Distinctive seasonal cycles in WAD2M can be observed across varying latitudinal bands. (Fig. 7). The Tropics (30°S-30°N)
contributes 68% of the global annual variation in wetland area, owing the large wetting and drying cycles of tropical wetlands. Despite its large area of intermittent wetlands, the mid-latitudes have a less pronounced seasonal cycle with an average annual minimum of 0.9 Mkm$^2$ and average annual maximum of 1.1 Mkm$^2$ compared to the tropics and high-latitudes. High latitude wetlands again have a strong seasonal cycle with an average annual minimum of 0.24 Mkm$^2$ and average annual maximum of 1.5 Mkm$^2$. The seasonal cycle of WAD2M in mid-latitude is small compared to GIEMS2
(Prigent et al., 2020), which is possibly due to different algorithms applied in SWAMPS and GIEMS2, especially in the way the vegetation contribution is accounted for. The seasonal cycle in the high latitudes is highest among the three regions, which is consistent with GIEMS2 and are mainly due to significant annual freeze/thaw cycle.

Given that there is a surprising scarcity of independent wetland products to evaluate the seasonal patterns in mid- and high-latitudes, we only focus on the comparison of seasonal cycle for the Amazon Basin, the largest regional contributors to the
seasonal cycle of wetland extent. For the Amazon Basin Lowlands, the estimates of wetland area exhibit a significant seasonal pattern in both the WAD2M and SAR-based high-resolution estimates from Hess et al. (2015). As illustrated in Fig. 8, the flooded/inundated wetlands in WAD2M vary considerably from 0.285 Mkm$^2$ in the low-water season (Oct.-Nov.) to 0.747 Mkm$^2$ in the high-water season (May-June). The average amplitude between the dry/wet seasons mapped by WAD2M is 0.461 Mkm$^2$, which is comparable to the estimated range 0.349 Mkm$^2$ from Hess et al., 2015 for the year 1995-1996. The
satellite products based on passive microwave bands such as SWAMPS, underestimate the seasonality and total wetland areas due to their limited ability to detect inundation outside of large wetlands and river floodplains. This indicates the needs to improve the retrieval approach to account for the vegetation contribution in the processing of active and passive microwave signal.



### 3.3.2 Interannual variation

The interannual variations in WAD2M suggests the effect of climate variations on global wetland extent across varying latitudinal bands (Fig. 7). Monthly anomalies, calculated by subtracting the 19-year mean monthly value from the monthly time series, reveal the changes in global wetlands in response to global climate variability such as the El Niño-Southern Oscillation (ENSO) (Fig 7b). For instance, a strong positive response in wetland areal anomalies was captured by WAD2M during the strong 2010-2011 La Niña event that temporarily increase the terrestrial water storage via affecting precipitation

patterns globally (Boening et al., 2012). The signal for the recovery captured by WAD2M, i.e., the decline during the late stage of La Niña, is consistent with the estimated terrestrial water storage from GRACE and the ESA CCI soil moisture product (Fig. 9). The linear fit of the pan-tropical wetland anomalies for WAD2M over 2000-2018 shows no significant change ($p > 0.1$) in the wetland extent for the entire period, consistent with (Parker et al., 2018) that showed no trend in tropical wetland emissions using satellite based inversion of $CH_4$ concentrations. Although the tropical regions have a net

reduction of $1.3 \times 10^3$ km/yr ($p < 0.05$) over the 2000-2018 period. There are no trends of wetland extent for mid-latitudes and high latitudes ($p > 0.1$) as was also found with Landsat imagery (Wulder et al., 2018).

In general, variation in surface water in the tropics is primarily driven by precipitation and the agreement in the patterns of the surface water extent and precipitation gives confidence in the inter-annual variability of wetland area estimation. At high latitude, surface-water runoff from snowmelt, not from direct precipitation, contributes towards the lower correlation

between inundation extent and precipitation. A strong decline in wetland area during the early stage of El Nino in 2015-2016 was captured by all of the products. The GIEMS2 dataset shows a similar patterns as WAD2M in two aspects: 1) Tropical wetlands contribute to over 50% of the global total wetland areas and the decadal change in wetland extent are mostly confined to the Tropics; 2) The temporal variations in WAD2M is consistent with GIEMS2, where a sharp decrease in the Tropics was found for 2010-2012, followed by a upward trend from 2013-2014. Note that despite the decline in 2000-2006,

the WAD2M estimate is followed by a slight recovery during the 2007-2014.

For the interannual variations at river basin scale, there is a generally good agreement in the interannual variation of wetland extent between WAD2M and four surface water products that are based on different methodologies (Fig. 9). All the products, including WAD2M, suggest a declining trend in wetland extent in Amazon Basin since 2012, with strong negative anomalies during 2015-2016 when the strong El Nino event occurred. The temporary increase in wetland extent in WAD2M

responding to the strong La Nina event of 2010-2011 is supported by the satellite-based observation of water storage and surface soil moisture, where good agreements of strong positive phase in terrestrial water were found for Orinoco River Basin and Paraná Basin in South America. The temporal variations of wetland extent for the Ganges River Nile, Yangtze, and Mekong River also suggest a good agreement between WAD2M and other products, except for the period since 2015 where GRACE observations suggest a strong decline in Ganges River Basin and Yangtze River Basin while WAD2M

remain constant or slightly declined. This discrepancy could be due to changes in irrigated rice extent, suggesting that the wetland extent in these regions so far is less influenced by groundwater depletion caused by human activity (Rodell et al.,





2018). This can be supported by the wetland extent estimate from the TOPMODEL based prognostic hydrological approach (Zhang et al., 2018), which explicitly exclude influence of human activity and attributes the change to the enhanced tropical precipitation since 2014.

## 415 3.4 Uncertainties in wetland areal estimation and future direction

Fig. 10 shows the uncertainty range (1σ) of mean annual maximum wetland area across the 6 global and regional data sources applied in this study. Amazon Lowland Basin and Siberian Lowland are two relatively more informed regions compared to the rest of the world (Fig. 10b). There is considerable uncertainty in wetland hotspots such as Hudson Bay Lowlands, West Siberian Lowlands, and major tropical floodplain regions. The causes of the high uncertainty for the boreal

and tropical wetlands differ. Mapping boreal wetlands requires discriminating between wetlands and small ponds, which are both considered as wetlands in some inventories (e.g., GLWD) but as inland waters in others (e.g., GSW). Thus, the removal of freshwater area is one reason that the boreal wetlands in WAD2M are lower. The uncertainty over tropical floodplain systems is due to the temporal mismatches of the different data sources, and the large seasonal and interannual variability in inundated area. Further, densely vegetated forest canopies in tropical floodplains can lead to systematic under-estimation of

inundation from satellite-based products. Also, uncertainty in DEMs (from spatial resolution, or whether the measurements are 'surface' or 'soil'), which serves as the basis of topographic index that is applied in the hybrid wetland mapping products (e.g. CIFOR, Tootchi et al., 2019), can lead to considerable uncertainty in estimation of wetland extent (Zhang et al., 2016), especially for the vegetated wetlands in complex terrain surface (Su et al., 2015).

## 4 Discussion

Due to the scarcity of ground-truth maps for representative regions, further work is needed to confirm the distribution of inundation captured by WAD2M representing an improvement over existing maps. In particular, the sensitivity to subcanopy inundation, the priori knowledge of land cover applied in the retrieval algorithm, and the length of observations can affect the overall accuracy of SWAMPS and thus contribute to the uncertainty of WAD2M. For instance, WAD2M reports a vast inundated area in the Sahel region where validation of the SWAMPS retrieval algorithm is lacking due to sparsity of

dynamic ground observations (Jensen and McDonald, 2019). Moreover, the decadal trends of WAD2M are influenced by the inter-calibration of brightness temperature across different microwave sensors, which could potentially introduce inconsistency between different time period covered by the measurements. Thus, it is important to be cautious with the interpretation of the long-term trends based on WAD2M. Lastly, because the GSW and MIRCA2000 data sources are aggregated to 0.25° spatial resolution in the processing of WAD2M, it ignores the potential overlapping between these two

mergers at fine spatial resolution, leading to unintentional double-accounting when deducting open water and rice paddies from WAD2M.





Future refinements to WAD2M could come from 1) improvements to revisit, spatial resolution, spectral range and signal-to-noise of remotely sensed data input and 2) refinements to our fusion methodology to use uncertainties to generate ensemble maps. Several new or upcoming satellite missions may provide improved global wetland dynamics in the future version of

WAD2M. The Cyclone Global Navigation Satellite System reflectometry (CYGNSS/GNSS-R) (Nghiem et al., 2017) demonstrate its capabilities to detect the inundation under different vegetation condition, which is complementary to inventories for evaluation. The NASA Surface Water and Ocean Topography (SWOT), the Copernicus L-band SAR mission Radar Observing System for Europe  (ROSE-L) (Pierdicca et al., 2019), and NASA-ISRO SAR (NISAR) mission, will greatly increase our capacity to monitor the spatiotemporal dynamics of wetlands and floodplains at high spatial resolution

(<50m), make it an immensely valuable resource in the future work of wetland dynamic mapping such as WAD2M. Commercial satellites are providing even higher-spatial resolution at daily revisit, i.e., PLANET Dove constellation, which is intercalibrated could go beyond providing static maps and provide time series of wetland data (Cooley et al., 2017).

For the methodology, combining products from different satellite sensors (e.g. optical and microwave) and inventories has been proved to be a feasible way to reduce the bias in the spatial distribution of wetlands and provide reliable estimates for

the use of global wetland $CH_4$ studies. However, the spatial resolution of WAD2M is dictated by the resolution of its input data on wetland dynamic dataset unless a downscaling methodology is applied. Downscaling can also be used to improve spatial resolution using artificial neural networks (see https://hess.copernicus.org/articles/22/5341/2018/hess-22-5341-2018-discussion.html) Machine learning approaches (Alemohammad et al., 2018; Kratzert et al., 2018; Wu et al., 2017) or physically-based hydrological models (Gumbricht, 2018), together with higher resolution images (e.g. Landsat, ALOS 1&2)

are better suited to capture inundation features at fine scales. On the other hand, inventories at the regional and national scales are needed for some less-informed wetlands (e.g. Africa, and Southeast Asia), which will help reliable validation and evaluation for these regions in the future quantitative studies of wetland. Moreover, even with better sensors in the future, improvements on wetland maps from past & future satellite will be necessary for backward extension of time series.

## 5 Data availability

The global wetland dynamic datasets in netcdf format is publicly available at https://doi.org/10.5281/zenodo.3998454 (Zhang et al., 2020).

## 6 Conclusion

The development of a global wetland product WAD2M has demonstrated the capability to produce maps of wetlands and inundation that are consistent with independent datasets. Combining temporal dynamics from coarse resolution product

SWAMPS with complementary products and inventories was shown to be a practical means of tracking variations of global wetland extent over time. WAD2M represents the most reliable representation of global vegetated wetland distribution to



date and will be useful to estimate wetland $CH_4$ flux. WAD2M provides valuable information for a range of applications, ranging from understanding the role of floodplains to carbon modelling and general assessment of global response to climate change.

## 475  Author Contributions

BP and ZZ conceived the work. All authors contributed to development of the wetland dataset, and analysis of results and writing of manuscript.

## Competing Interests

The authors declare that they have no conflict of interest

## 480  Acknowledgments

This study was supported by the Gordon and Betty Moore Foundation through Grant GBMF5439 "Advancing Understanding of the Global Methane Cycle" to University of Maryland supporting the Methane Budget activity for the Global Carbon Project (globalcarbonproject.org). K. Jensen's effort was supported through an award from the NASA Earth and Space Science Fellowship Program under Grant 80NSSC17K0387. Development of the SWAMPS dataset was supported by the NASA Making Earth System Data Records for Use in Research Environments Program under Cooperative Agreement NNX11AQ39G and Cooperative Agreement NNX11AP26A. Portions of the work were carried out at the Jet Propulsion Laboratory, California Institute of Technology, under contract to the National Aeronautics and Space Administration. A. Bartsch's, G. Hugelius´s, and T. Gumbrichts, contribution is supported through the Nunataryuk project, funded under the European Union's Horizon 2020 Research and Innovation Programme under grant agreement no. 77342. The wetland mapping was partly ran in cooperation with the Center for International Forestry Research (CIFOR) and its SWAMP programme (Sustainable Wetlands Adaptation and Mitigation Programme), with support from the U.S. Agency for International Development (USAID) Grant #MTO069018. The development of global models for geomorphology and hydrology was partly done with support from the World Agroforestry Centre (ICRAF). We would also like to acknowledge the CGIAR research programmes on Forests, Trees and Agroforestry (FTA) and the Climate Change, Agriculture and Food Security Programme (CCAFs).



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





**Tables**

**Table 1: Three states of inundation (Unit: $10^3$ km$^2$) at different steps of the processing of WAD2M.**

| State of inundation | Continent | Original SWAMPS | After wetland upward merger | After removal of rice & open water (WAD2M) |
|---|---|---|---|---|
| MA$_{min}$ | Africa | 499.4 | 641.6 | 456.7 |
| | Asia | 1,387.8 | 1,685.8 | 1043.9 |
| | Europe | 366.0 | 391.3 | 202.7 |
| | Central & South America | 610.2 | 943.2 | 721.8 |
| | North America | 1,421.5 | 1,724.1 | 892.2 |
| | Oceania | 206.7 | 212.0 | 189.0 |
| | **Global** | **4,491.7** | **5,598.0** | **3,506.4** |
| MA$_{max}$ | Africa | 1,021.7 | 1,252.5 | 1,057.8 |
| | Asia | 3,018.5 | 3,692.0 | 2618.3 |
| | Europe | 851.5 | 907.5 | 605.8 |
| | Central & South America | 803.3 | 1,323.2 | 1097.7 |
| | North America | 2,229.4 | 2,820.8 | 1,799.5 |
| | Oceania | 336.1 | 348.6 | 331.3 |
| | **Global** | **8,260.5** | **10,344.6** | **7,510.4** |
| MA$_{Lt}$ | Africa | 1,726.3 | 2,151.8 | 1729.4 |
| | Asia | 4,832.1 | 5,947.5 | 4523.0 |
| | Europe | 1,534.7 | 1,624.1 | 1,234.1 |
| | Central & South | 1,237.7 | 2,238.1 | 1818.2 |





| America | | | |
|---|---|---|---|
| North America | 3,274.5 | 4,222.2 | 2993.9 |
| Oceania | 787.2 | 821.0 | 721.2 |
| **Global** | **13,392.7** | **17,004.7** | 1,3020.0 |





**Table 2: Summary of wetland areas (Mkm²) in WAD2M by latitudinal bands in comparison with the long-term maximum wetlands from merger datasets applied in the WAD2M processing and independent evaluation datasets.**

| Wetland Datasets | | | Latitudinal Bands | | | |
|---|---|---|---|---|---|---|
| | | Metric | 60°N-90°N | 30°N-60°N | 90°S-30°N | Global |
| SWAMPS[*] | | $MA_{min}$ | 0.7 | 1.4 | 2.1 | 4.2 |
| | | $MA_{max}$ | 1.6 | 3.3 | 3.6 | 8.5 |
| | | $MA_{Lt}$ | 2.5 | 5.1 | 5.6 | 13.4 |
| Static merger datasets | GSW[†] | $MA_{Lt}$ | 1.1 | 2 | 1.4 | 4.5 |
| | CIFOR[*] | $MA_{Lt}$ | NA | 0.5 | 4.1 | 4.6 |
| | NCSCD& CAWASAR[‡] | $MA_{Lt}$ | 2.3 | 0.9 | NA | 3.2 |
| | GLWD[*§] | $MA_{Lt}$ | 1.5 | 3.1 | 4.2 | 8.8 |
| | MIRCA2000[¶] | $MA_{Lt}$ | 0 | 0.1 | 0.3 | 0.4 |
| WAD2M[#] | | $MA_{min}$ | 0.5 | 1.1 | 1.7. | 3.5 |
| | | $MA_{max}$ | 1.2 | 3.1 | 3.2 | 7.5 |
| | | $MA_{Lt}$ | 2.1 | 5.4 | 5.5 | 13.0 |
| Comparison datasets | GIEMS2[*] | $MA_{Lt}$ | 1.9 | 3.9 | 5.9 | 11.7 |
| | Tootchi2019[**] | $MA_{Lt}$ | 3.6 | 8.4 | 16.8 | 28.8 |

*represents inundated area.

†represents open water.

‡represents peatlands and mineral wetlands.

§GLWD excludes rivers, lakes, and reservoirs; wetland classes interpreted to be at middle range.

¶ represents irrigated rice paddies.

#includes both inundated and non-inundated wetlands but excludes artificial inundation and lakes, ponds, and reservoirs.

** For Tootchi et al., 2019, we use the CW-TCI version and exclude lake areas.





## Figures

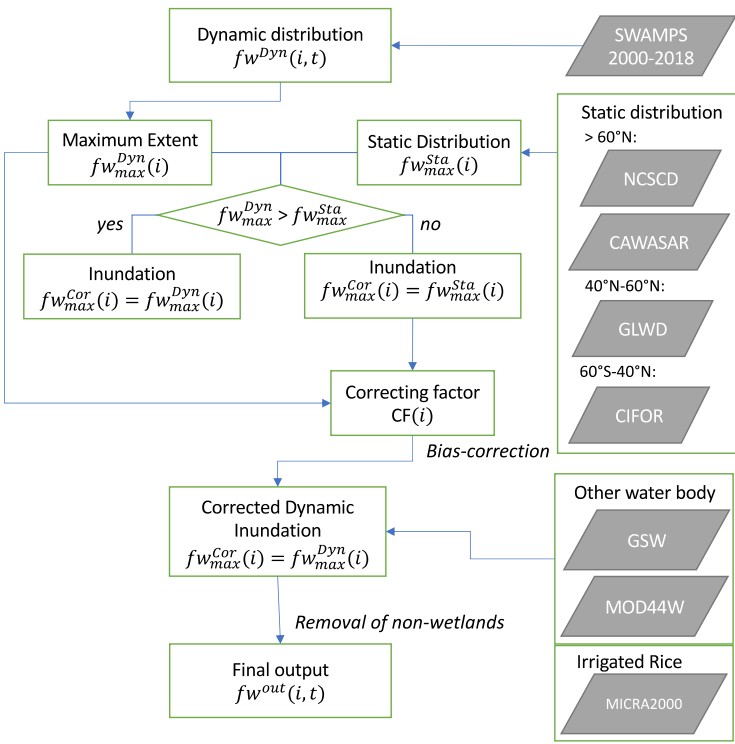

**Figure 1: Flow chart of the method describing the wetland extent estimate from SWAMPS and other water body datasets to consolidate our WAD2M product. The chart describes the step calculating fw in each 0.25° grid-cell i at time (monthly) t.**




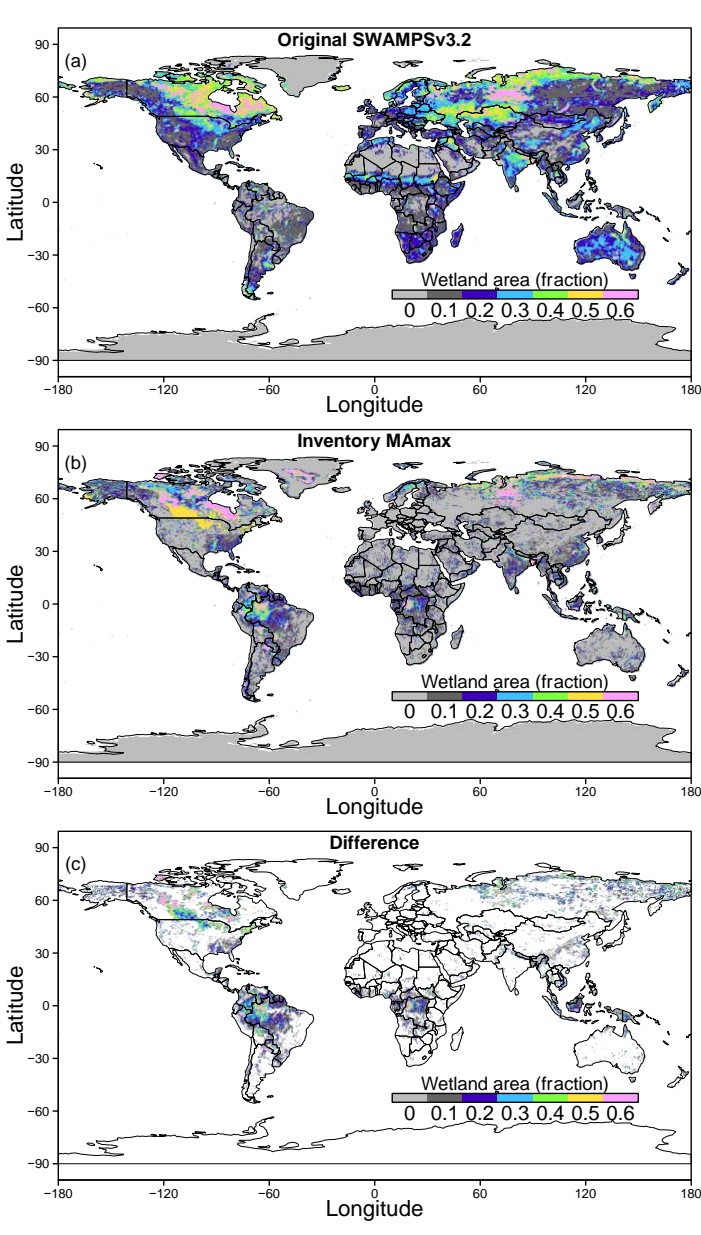



**Figure 2: Global maps of wetland fraction before and after processing for (a) global distribution of MA$_{max}$ of SWAMPS; (b) global distribution of MA$_{max}$ superimposed by NCSCD, GLWD, and CIFOR, (c) difference of MA$_{max}$ between inventories and original SWAMPS (inventories minus SWAMPS); Here the only the positive difference is shown as only the regions with positive values apply the upward correcting factors.**




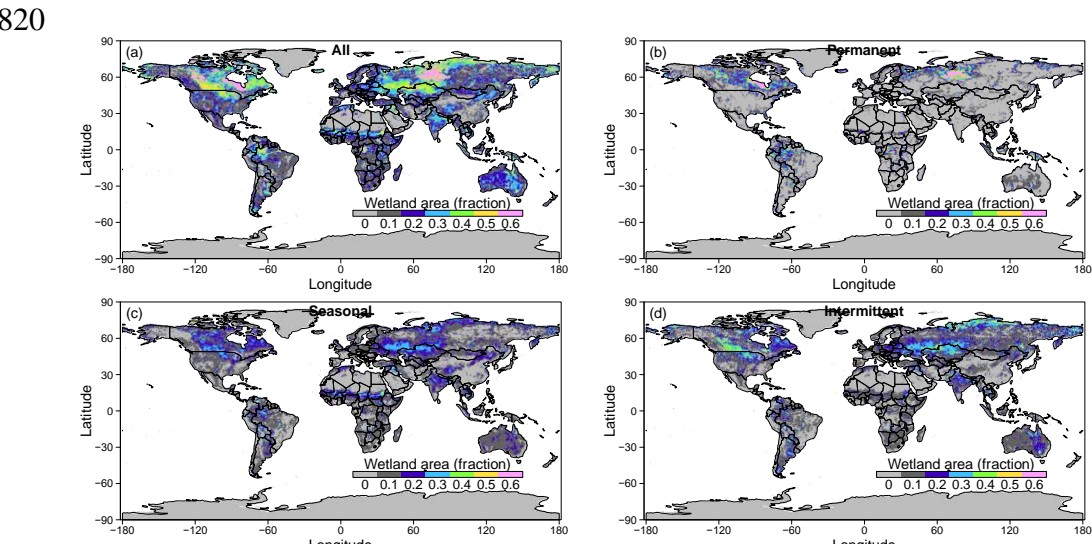

**Figure 3: Maps of wetland extent in WAD2M for (a) long-term maximum ($MA_{Lt}$); (b) mean annual minimum ($MA_{min}$); (c) mean annual magnitude ($MA_{max}$ minus $MA_{min}$); and (d) intermittent inundated ($MA_{Lt}$ minus $MA_{max}$).**




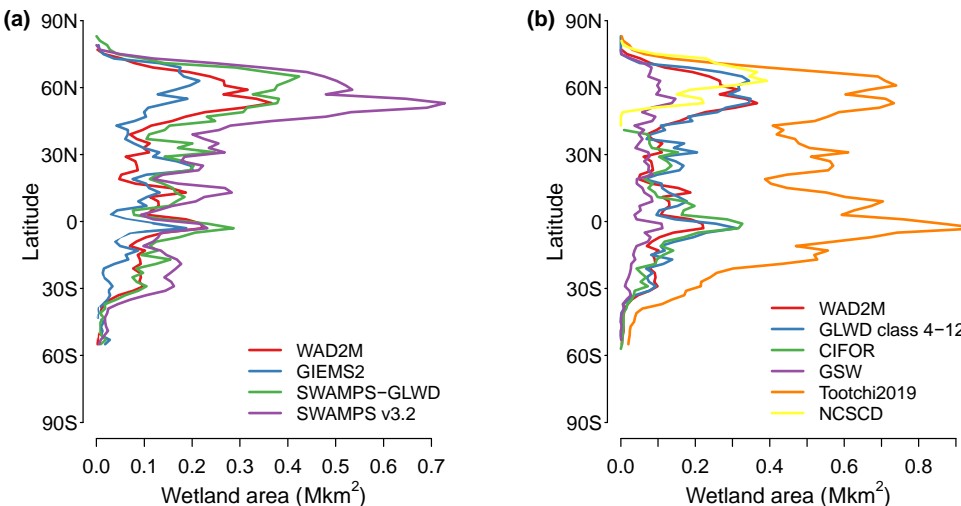

**Figure 4: Latitudinal gradient of WAD2M in comparison to existing wetland/inundation products. (a).**
**Comparison with the dynamic inundation products GIEMS2 and SWAMPS-GLWD (previous version of**
**WAD2M). The solid lines represent long-term mean annual maximum (MA$_{max}$) with upper (MA$_{Lt}$) and**
**lower (MA$_{min}$) range of inundated area marked as shaded area. (b), Comparison with static inventory**
**datasets. To make it a fair comparison, lakes, rivers, and reservoirs (i.e. class 1-3 in GLWD) were**
**excluded from GLWD. Lakes (code 4) were excluded from topographic index (TCI) version of the global**
**wetland composite map from Tootchi et al., 2019 (denoted as Tootchi2019). The wetland area is**
**calculated by 2-degree latitudinal band.**

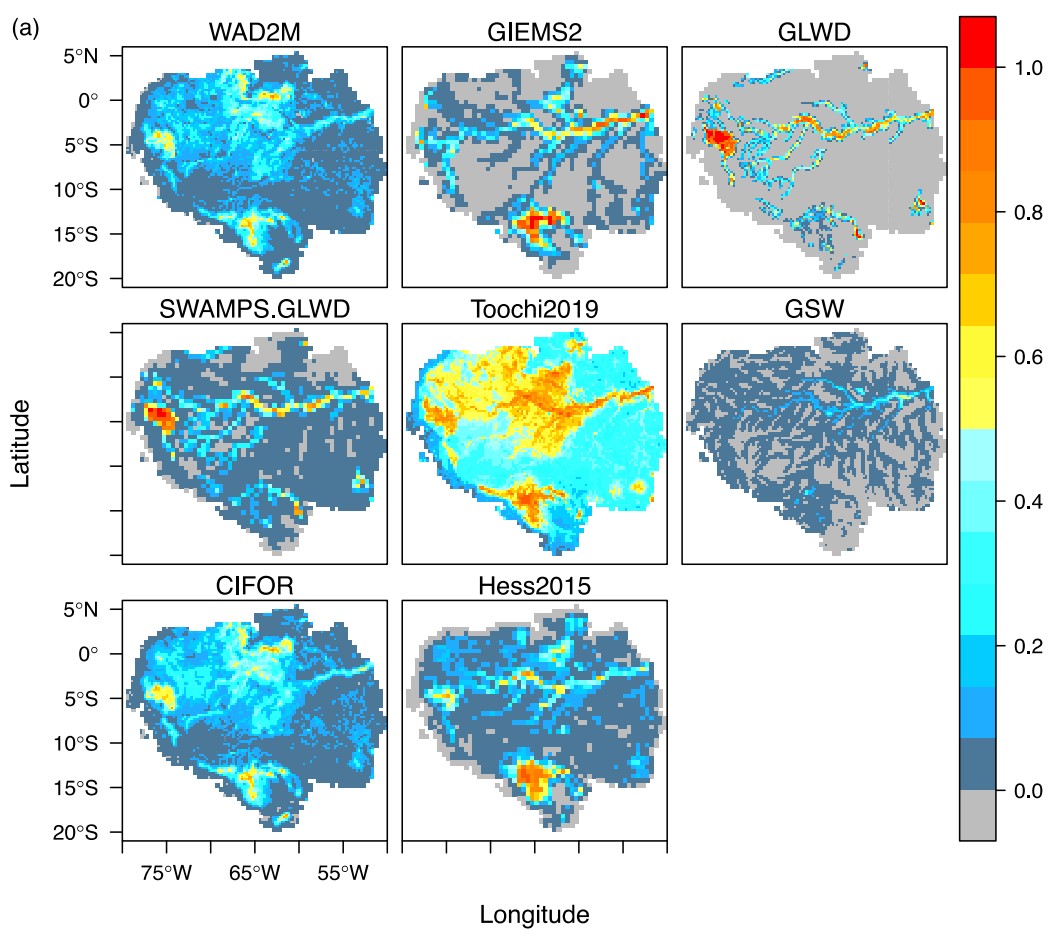

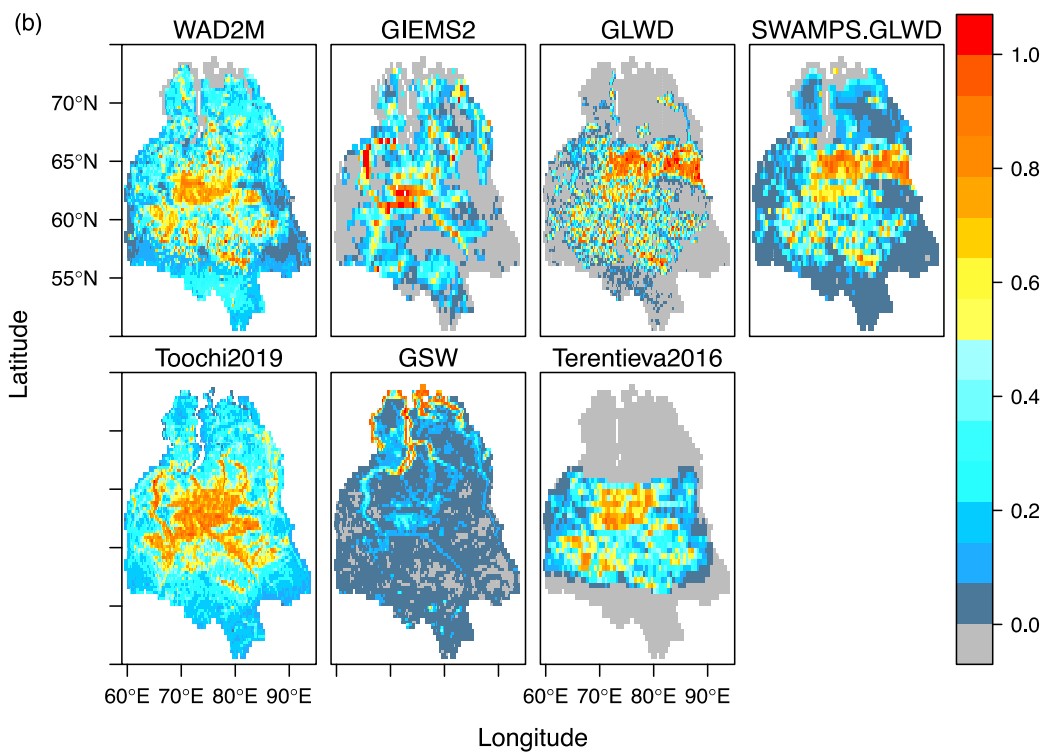

**Figure 5: Maps of fractional wetland area from WAD2M in comparison with benchmark datasets for (a) Amazon Basin Lowland; (b) West Siberian Lowland. The wetland maps from WAD2M, GIEMS2, GSW, and CIFOR represent long-term maximum while the fractional inundation for Amazon Lowland from Hess et al.,2015 represents wetland during the period 1995-1996 for the high-water season. GLWD represents GLWD level 3 that excludes lakes and reservoirs.**

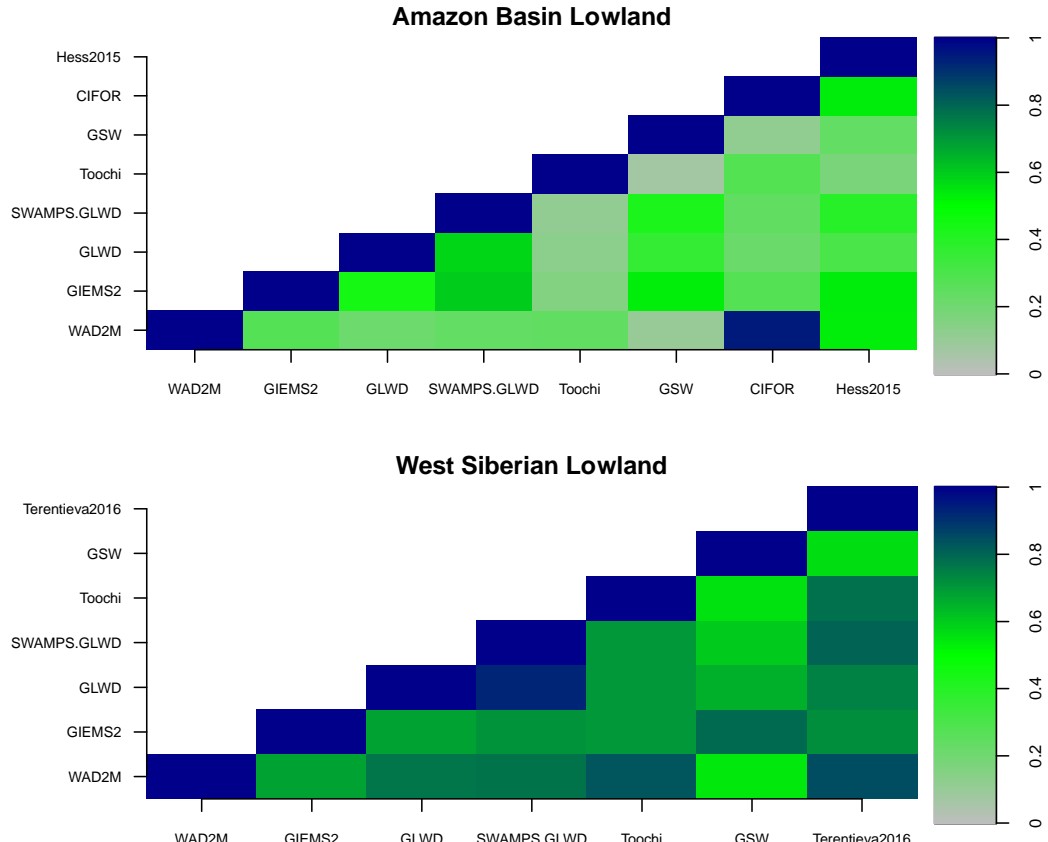

**Figure 6: Matrix of Kappa coefficient for each pairwise comparison for (a) Amazon Lowland Basin and (b) West Siberian Lowland. The Hess et al., 2015 and Terentieva et al., 2016 represent two independent regional datasets. All of the datasets in (b) were masked by the map of Terentieva et al., 2016, which is**




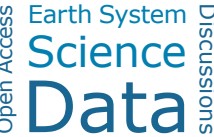

available for the taiga forest zone, hence the calculation of the Kappa coefficient excludes the arctic tundra zone (latitude > 65°N).

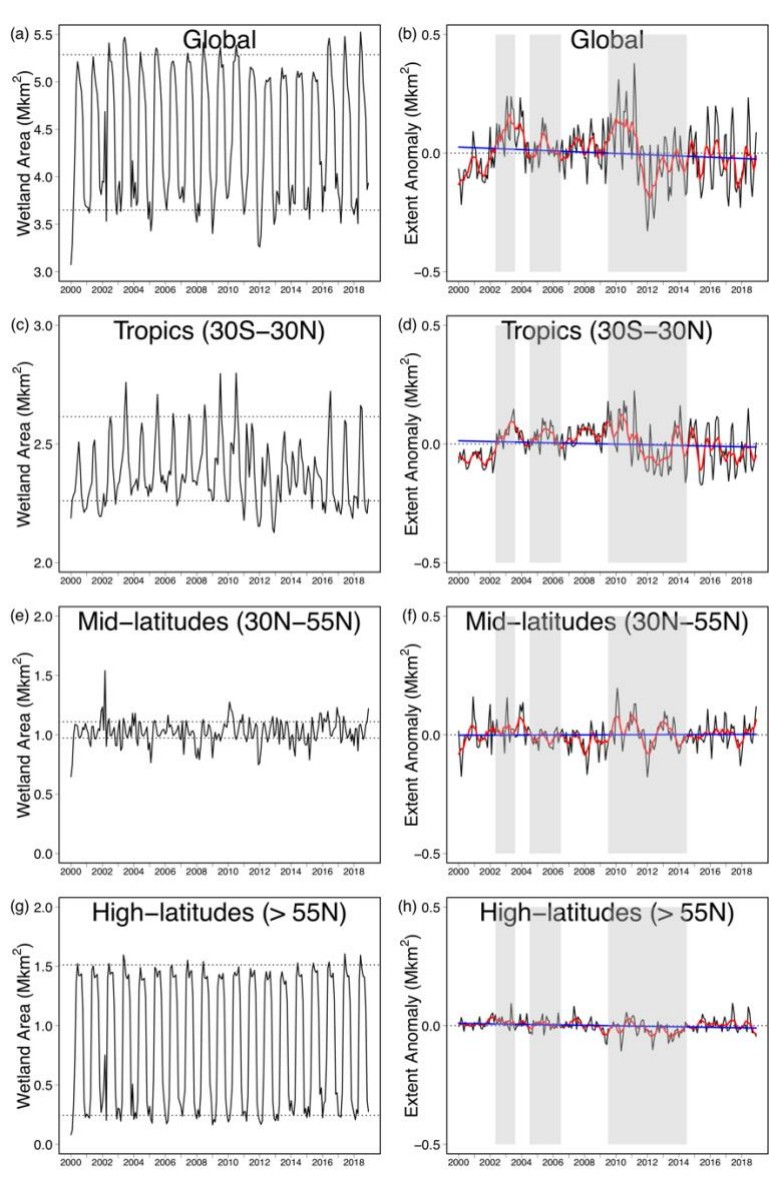





**Figure 7: The WAD2M wetland extent and their anomalies from 2000-2018. Monthly-mean wetland extent (left column; a,c,e,g) for 2000-2018 in black, for the globe, Tropics (30°S-30°N), and mid-latitudes (30°N-55°N) and northern high latitudes (latitudes > 55°N). Horizontal dashed lines in the left column panels represent the mean annual maximum and the mean annual minimum. In the right column (b.d,f,h), the deseasonalized anomalies (black) with the 6-month running mean (red) and linearly fitted trends**
**(blue) using least-squares regression were listed. Shaded areas represent the La Niña phase from NOAA multivariate ENSO index.**



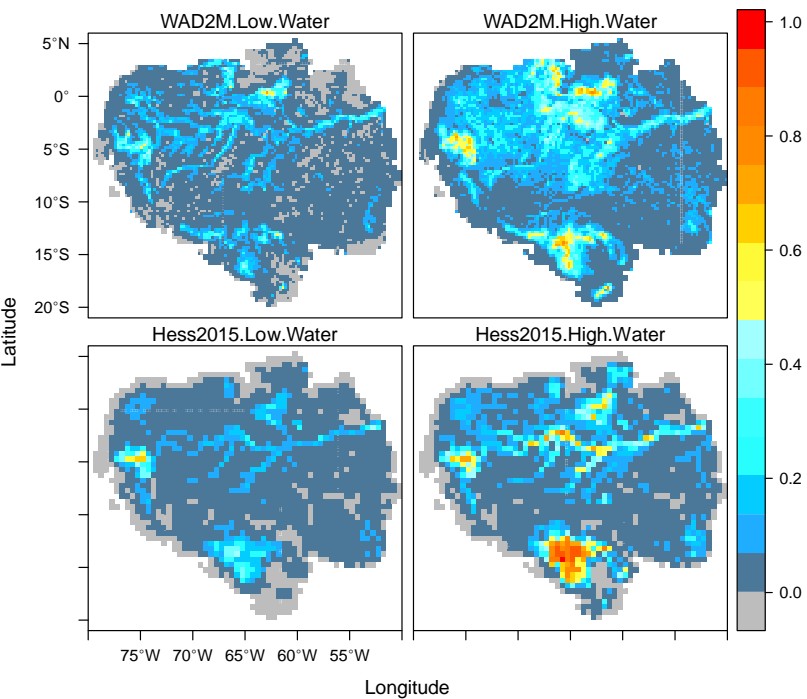

**Figure 8: Spatial distribution of wetland areal fraction during the low- (Oct-Nov) and high-water (May-July) seasons for Amazon Basin lowland at 0.25 degree cells.**

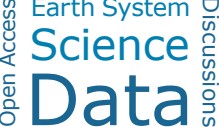

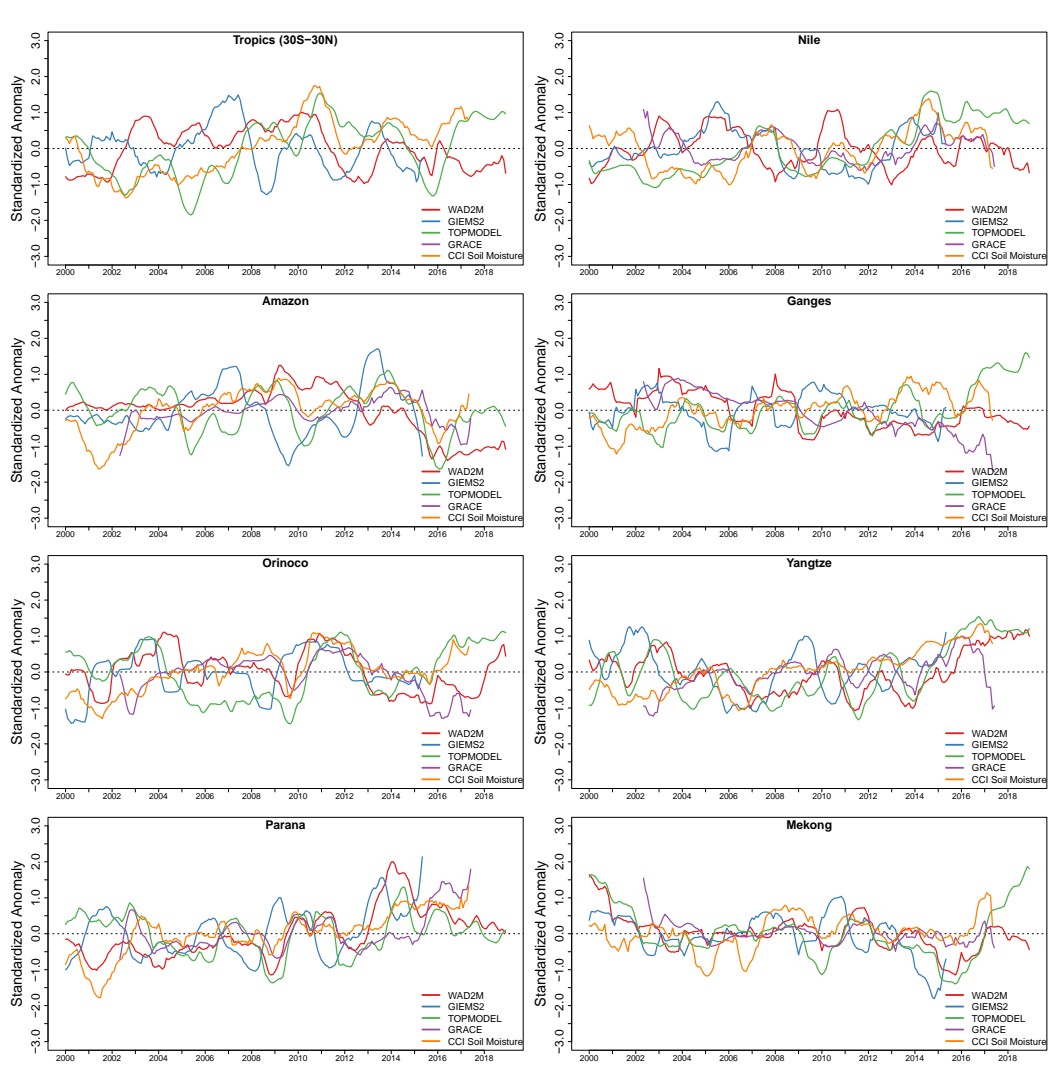





**Figure 9: Temporal variations of wetland anomalies in WAD2M in comparison with terrestrial water products from multiple sources. The anomaly values were standardized using the Z-score. The shaded lines and the solid lines represent standardized value and the 12-month running mean of the anomalies.**



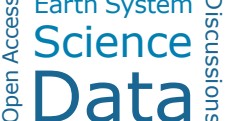

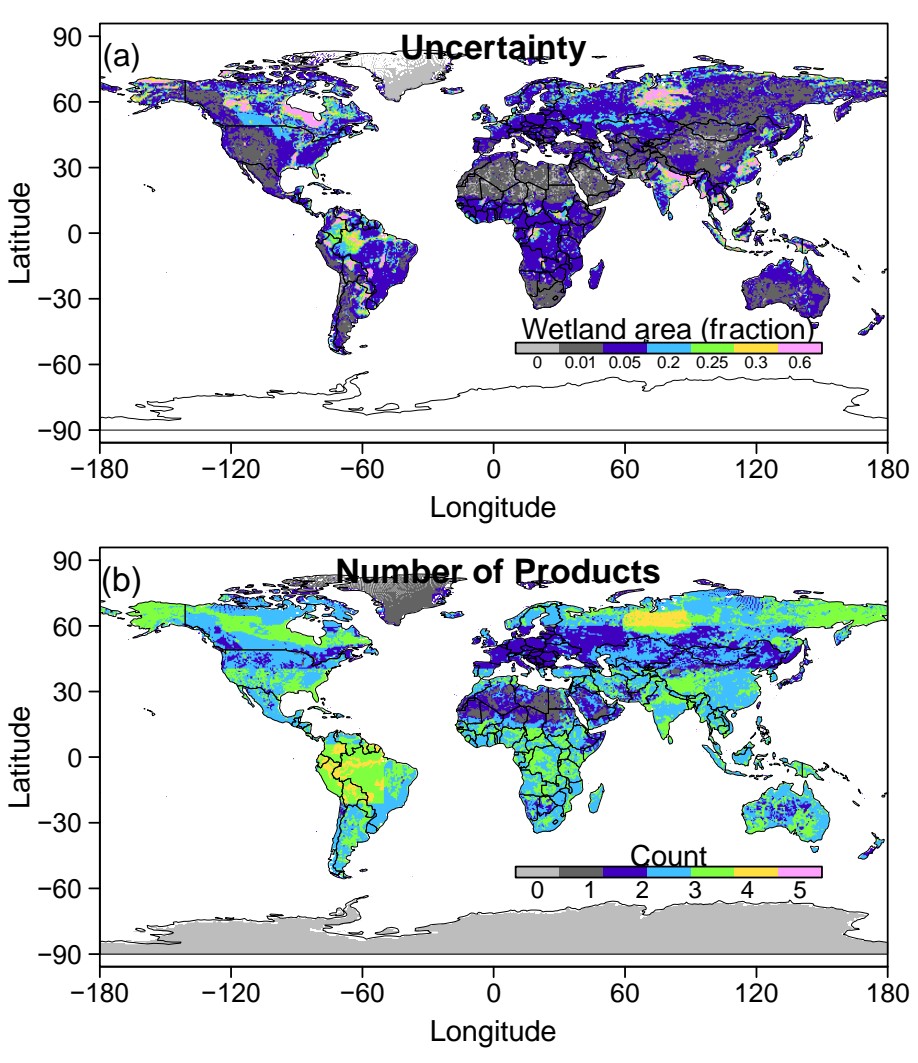



**Figure 10: Uncertainty (1-σ) of annual maximum wetland area (MA_max) across data sources. The data**
**sources used in the calculation are WAD2M, GIEMS2, GLWD, CIFOR&CAWASAR, NCSCD,**
**Toochi2019, Terentieva et al., (2016), and Hess et al., 2015. (a) spatial distribution of uncertainty in**
**wetland fraction. (b) Number of products used in the calculation.**



**Appendix A**

**Table A1. Overview of existing global and regional datasets of open water, wetland, and related proxy datasets.**

| Name and reference | Resolution | Type | Source | Spatial coverage | Temporal coverage |
|---|---|---|---|---|---|
| Global | | | | | |
| Matthews and Fung, (1987) | 1° | Wetlands | Digital maps for vegetation, soil, and inundation | Global | Static |
| GLWD-3 (Lehner and Döll, 2004) | 30 arcsec, ~ 1km | Inland water bodies (lakes, reservoirs and wetlands) | Digital database based on inventories | Global | Static |
| GIEMS (Prigent et al., 2007) | 0.25° | Inundation | Multiple satellite fusion | Global | 1993-2007 |
| GIEMS2 (Prigent et al., 2020) | 0.25° | Inundation | Multiple satellite fusion | Global | 1992-2015 |
| GIEMS-D15 (Fluet-Chouinard et al., 2015) | 15 arcsec, ~500 m | Inundation | Multiple microwave sensors | Global | 1993-2007 |
| GIEMS-D3 (Aires et al., 2017) | 90 m | Inundation | Multiple satellite fusion | Global | 1993-2007 |



| | | | | | |
|---|---|---|---|---|---|
| Feng et al., (2016) | 30 m | Surface water | Landsat images | Global | Static |
| G2WBM (Yamazaki et al., 2015) | 3 arcsec, ~ 90 m | Surface water | Landsat images | Global | Static |
| GLOWABO (Verpoorter et al., 2014) | 14.25 m | Lakes | Inventories | Global | static |
| HydroLakes (Messager et al., 2016) | 15 arcsec, ~ 500 m | Lakes | Geo-statistical approach based on topographic data and inventories | Global | static |
| Tootchi et al., (2019) | 15 arcsec, ~ 500 m | Wetland composites | Hybrid of satellite imagery and groundwater modeling | Global | static |
| GFPLAIN (Nardi et al., 2019) | 250 m | Floodplain | Hydrological Model | Global | static |
| MOD44W (Carroll and Loboda, 2017) | 30 m | Surface water | Landsat images | Global | static |
| GRWL (Allen and Pavelsky, 2018) | 30 m | Rivers and streams | Hybrid of in situ measurements and Landsat images | Global | static |
| GSW (Pekel et al., 2016) | 30 m | Surface water | Landsat images | Global | 1980-2016 |
| (Wu et al., 2017) | 0.5° | Peatlands | Machine learning based on climate, soil and topographic datasets | Global | static |
| SWAMPS (Jensen and McDonald, 2019) | 0.25° | Inundation | Multiple microwave images | Global | 1992-2018 |





| SWAMPS-GLWD (Poulter et al., 2017) | 0.5° | Wetlands | Hybrid of satellite products | Global | 2000-2012 |
| PEATMAP (Xu et al., 2018) | Polygon | Peatlands | Meta-analysis based on various sources | Global | static |
| MIRCA200 (Portmann et al., 2010) | 5 arcmin | Irrigated Rice paddies | Inventories | Global | static ca. 2000 |
| (Du et al., 2016) | 5 km | Open water | AMSR-E and MODIS | Global | 2002-2011 |
| GRACE (Landerer and Swenson, 2012) | 1° | Land water mass equivalent | GRACE gravity satellite | Global | 2003-2012 |
| Yan et al., (2017) | Polygons | Wetland complex | Inventories | China | static |
| Zhang et al., (2016) | 0.5° | Wetlands | Hydrological model | Global | 1980-2017 |
| ESA SMOS (Kerr et al., 2012) | 0.25° | Soil moisture | Microwave images | Global | 2011-2017 |
| SMAP (Reichle, 2018) | 9 km | Soil moisture | Microwave images | Global | 2015-2018 |
| Regional | | | | | |
| Hess et al., (2015) | 100 m | Floodplain | SAR JRES-1 | Amazonia | Static |
| NCSCD (Hugelius et al., 2013) | 1 km | Permafrost peatlands | Polygon-based digital inventories | Pan-Arctic (> 45°N) | Static |
| Wulder et al., (2018) | 30 m | Wetlands (non-treed and treed | Landsat land cover maps | Canada | 1984-2016 |





| | | | | | |
|---|---|---|---|---|---|
| | | combined) | | | |
| Amani et al., (2019) | 30 m | Wetlands (bog, fen, marsh, swamp, and shallow water) | Landsat images | Canada | Static |
| Li et al., (2019) | 500 m | Surface water | MODIS images | Mediterranean region | 2000-2017 |
| Jin et al., (2017) | 90 m | Wetland composites | Landsat and LiDAR | Delmarva Peninsula | 1985-2011 |
| DeVries et al., (2017) | 30 m | Wetland composites | Landsat images | North America | static |
| CIFOR (Gumbricht et al., 2017) | 232 m | Wetlands | Expert system approach based on topography, soil, and climate datasets | Pan-Tropical | static |
| PeRL (Muster et al., 2017) | Polygons | Ponds and lakes | Optical aerial and satellite imagery | Circum-Arctic | static |
| Terentieva et al., (2016) | 30 m | West Siberian Lowland | Landsat validated by field data | West Siberian Lowland | static |
| CAWASAR (Widhalm et al., (2015) | 500m | Arctic tundra wetlands | ENVISAT ASAR Images | Circum-Arctic | 2005-2011 |