# Peer review of "Development of a global dataset of Wetland Area and Dynamics for Methane Modeling (WAD2M)"

_Earth System Science Data, 2020_

## Referee Comment (RC1) · Anonymous Referee #1 · 15 Jan 2021

Dynamic wetland datasets are no doubt essential in monitoring and estimating global methane budget. The efforts made by the authors to combine available wetland datasets including those from remote sensing, ground survey and modelling are comprehensive and valuable. There are, however, a few concerns related to the rationale of the data fusion and comparison results as follows:

(a) A better justification of "fwmax to match the wetland maps for pixels where fwmax is less than the static distribution" in the data fusion is needed. According to Schroeder et al., 2015, SWAMPS retrievals represent "water surface within open areas and under low density vegetation" due to the relatively low penetration ability of the microwave frequency used. Therefore, it is likely that SWAMPS will have overall underestimated water fraction for vegetated areas. In addition, both SWAMPS and the

static water datasets have biases, uncertainties and inconsistency in their representative season/periods. Ideally, all these factors should have been carefully accounted for when fusing the datasets, though it might not be practically achievable. The information of relative changes of SWAMPS water area seems more important to me than the absolute values when merging the datasets.

(b) The use of GSW to identify and mask out inland open water bodies seems over-simple to me. Assuming a lake with seasonal inundation changes, the pixels detected as water for less than 50% of the months were not classified as inland water (section 2.2.2), but they may be part of the lake over the wet season.

(c) What caused the overestimation of water fraction in dry areas such as central Australia (Fig.2a, 3a)? Did the uncertainty associated with drylands also affect wetland areas?

(d) Is it possible to examine data quality and accuracy for an area where ground/aircraft-based wetland mapping is available?

Minor comments: (a) For Fig. 9: please provide statistics on correlations between WAD2M times series and the others.

(b) For Fig. 9: did you miss GRACE time series in the upper-left figure?

(c) Line 91-92: please revise the sentence, which is not accurate.

(d) Line 101-102: please revise to improve clarity.

(e) Line 104: did you mean "inundation under snow"? Any reference to support this?

(f) Line 193-194: "The coastline..." Not sure about the meaning. Please revise or clarify.

---

## Referee Comment (RC2) · Joe Melton (Referee) · 12 Feb 2021

Review of Zhang et al. (ESSD)

Zhang et al. generate a novel dataset of global wetland dynamics using the remotely-sensed dataset, SWAMPS, as a starting point. They then address its shortcomings by bringing in regional datasets for known issues like saturated wetlands in the high latitudes (e.g. using NCSCD) or trouble seeing through canopy in tropics (e.g. using CIFOR). I generally find what they have done to be quite logical. I think the suggestion of an ensemble approach for future versions would be very useful and was glad to see it suggested. At present, the approach here relies quite heavily on the performance of SWAMPS/GLWD/NCSCD/CIFOR being the best products out there. That is unlikely

to be correct for all locations so more regional specific datasets could likely improve future versions of WAD2M. My biggest complaint about the paper was the graphics and specifically the colour schemes. This sounds trivial but it did make interpreting the figures challenging. I hope the authors take my comments there seriously. As I think my comments can be relatively easily dealt with I am suggesting minor revisions.

- line 51: also conversion to rice agriculture?

- l57: does Ramsar include rivers/lakes/ponds as wetlands? That is how I read that but am not sure if that was what was intended.

- l65: Can you provide an example of a dataset that omits them?

- l.91 - detect flooding beneath most vegetation canopies - this begs the question which canopies can it then not detect the flooding underneath?

- l.92 - maybe specify the frequency you are talking about, could be construed to mean satellite return frequency or something rather than frequency in the electomagnetic spectrum, esp as the next sentence talks about temporal coverage.

- l. 104 - Is there any issue with high-latitude regions due to the satellite orbits or the low radiation in the winter that should be mentioned here?

- l. 143 - the problem with this approach is that you then end up potentially increasing the wetland area since you are assuming that SWAMPS is wrong and the wetland product is correct. Given how error prone large-scale wetland mapping can be I think this is likely a dangerous assumption. I don't really disagree with the approach though, as there is not likely a better way to include wetlands that SWAMPS doesn't see. I would suggest that somewhere it is noted that this approach could inflate wetland areas as it would then include wetlands that may be erroneous in the wetland inventory.

- l. 145 - Hugelius has published a new effort for high latitude peatlands, while I understand that this work might have been completed prior to the publication of that work, I wonder if there is much change between the NCSCD than the newer paper (Hugelius

et al. 2020)? I see Hugelius is on the author list.

- l 149 - I wonder if the dataset used for the temperate regions could be improved upon by using updated datasets for more specific regions. E.g. for Canada (Mahdianpari et al. 2020). Since the wetland inventory seems important (my comment for line 143), it seems reasonable to try to get the very best inventories. I am not sure if GLWD is that (although it was a good first attempt).

- l. 181 - what proportion of SWAMPS 3.2. observations were 'valid'? How many valid observations were needed to be able to produce the monthly values?

- l. 183 - What does it mean that they were later 'refiltered'?

- l. 186 - When I tried to use the GSW for a project I found it to have large issues with stripping of the data due to satellite passes and data quality control. Was this problem also encountered and if so how was the influence of the stripping minimized?

- Also with GSW, since it is based on Landsat does this mean it is also just the open waterbodies (i.e. no canopy cover)? I think you say this on line 196 but not earlier in describing what GSW provides so I want to be sure.

- L 211: I think this is getting to my question about how it compares to the newer Hugelius product.

- L. 230 to get an idea of how good the datasets agree for areas of overlap, was any attempt made to see how CIFOR and GLWD compare against each other or NCSCD vs. GLWD? If the agreement is poor then it doesn't allow a lot of confidence in the use of these datasets.

- Table 1 - misplaced comma final column and row.

- It would be useful to explain why SWAMPS was the starting remotely-sensed dataset and not GIEMS. Why one over the other? Assumedly there was some performance improvement, like is seen in Fig 4a where SWAMPS seems more sensitive to high

latitude wetlands?

- In Fig 1 the Inundation box dead-ends. So that information is not integrated into the final output? I think a line is missing. Also isn't the correcting factor applied to dynamic inundation, not becoming the corrected dynamic inundation as is shown? This figure is really hard to follow. I suggest a re-think.

- The maps have a, creative, colour scheme. Have you considered a linear colour map? There are good resource around like, https://sciviscolor.org/. At present there is an arbitrary strong visual cut-off at between 0.1 and 0.2 giving some sort of importance to a threshold that has none. Is the maximum value in all of the dataset 0.6? Also the maps could have Antartica cut off to allow a larger representation of the non-ice covered parts of the world. Also it would be nice if the images could have improved resolution. When I tried to zoom into see the distribution it looks as though the plotting used a setting that interpolated between gridcells, i.e. pixels are washed out. Please check and correct this. BTW, Fig 10 with the number of sources panel is the only time I think the colour scheme makes sense. Here it is categorical so the use of different colours with no continuous colour scheme makes perfect sense (small point on that one - why have a 0 value? Isn't that only for Antarctica which should be trimmed anyway?).

- Fig 5, there is a hashtag you might want to check out #endrainbow. It is based on the fact that rainbow colour scheme tends to distort the perception of the data since values (e.g. yellows) with no greater importance stand out more to readers, but shouldn't as there is nothing important about the 0.5-0.6 fraction. Please consider using colour schemes that follow best practices for data visualization.

- Fig 5 when the grey value is below 0 on the colour bar, how do we interpret that? It is missing/no values or 0? I assume it must be missing/no since Ternetieva isn't really showing wetlands then none for the most S and N part of the domain. But then other plots have the 0/missing/no wetland grey in the middle of their domains, so what is going on?

[Figure]

- L 360 - 'owing to'

- Fig 6 - Sorry to continually complain about colour schemes but why not just use a linear blue colour scale? Why add in the green? It makes me think there is something especially important about middle of the road values, which isn't the case, so please reconsider. E.g. the most visually striking box in the lower plot (WSL) is for WAD2M and GSW - is that really the thing you want to stick in the reader's mind?

- Fig 8 over the same years? What years are they?

- Fig 9 the caption doesn't mention if these are the watershed values and how that might be defined. Might be worth including a small sub-map showing the boundaries.

- l388 - I think the comparison with Parker et al. 2018 is a bit tenuous, wetland extent is only part of what affects CH4 emissions so the consistency could just be fortuitous. Also this confuses me a bit, it says no significant trend for pan-tropical then the next sentence says tropical have net reduction. So how does pan-tropical differ from tropical?

- L430- this sentence seems to be saying two things: 1) further work needed to confirm and 2) improvement over existing. I suggest rewording so it is more one thought rather than a mash of two that conflict.

- L443 - I quite like this point about ensemble maps. I think it would make version 2.0 much better.

- L457 - artificial neural networks are a machine learning approach.

- L461 - what are less informed wetlands?

- Some regions may be 'not wetland' then become wetland for, say a month since that is the shortest time this dataset can do, is the assumption that the system can behave like a wetland in that amount of time? Since the models will run their wetland model over this as a mask I think that is an underlying assumption but I am not sure if it is

true. This is more of a comment than question and I am sure that number of pixels like that are minimal but I think the assumption is made that each time a pixel 'becomes' a wetland it is assumed to immediately adopt a wetland behaviour from a biogeochemical perspective. This is likely out of the scope of this paper but it does play into how this dataset is used.

Refs cited:

Hugelius, G., Loisel, J., Chadburn, S., Jackson, R. B., Jones, M., MacDonald, G., Marushchak, M., Olefeldt, D., Packalen, M., Siewert, M. B., Treat, C., Turetsky, M., Voigt, C. and Yu, Z.: Large stocks of peatland carbon and nitrogen are vulnerable to permafrost thaw, Proc. Natl. Acad. Sci. U. S. A., 117(34), 20438–20446, 2020.

Mahdianpari, M., Salehi, B., Mohammadimanesh, F., Brisco, B., Homayouni, S., Gill, E., DeLancey, E. R. and Bourgeau-Chavez, L.: Big Data for a Big Country: The First Generation of Canadian Wetland Inventory Map at a Spatial Resolution of 10-m Using Sentinel-1 and Sentinel-2 Data on the Google Earth Engine Cloud Computing Platform, Can. J. Remote Sens., 1–19, 2020.

---

## Author Comment (AC1) · 30 Mar 2021

Dynamic wetland datasets are no doubt essential in monitoring and estimating global methane budget. The efforts made by the authors to combine available wetland datasets including those from remote sensing, ground survey and modelling are com- prehensive and valuable. There are, however, a few concerns related to the rationale of the data fusion and comparison results as follows:

We thank the reviewer for the interest in our study and for the constructive and insightful comments that help us to improve our manuscript. Each comment is addressed below in detail.

(a) A better justification of "fwmax to match the wetland maps for pixels where fwmax is less than the static distribution" in the data fusion is needed. According to Schroeder et al., 2015, SWAMPS retrievals represent "water surface within open areas and under low density vegetation" due to the relatively low penetration ability of the microwave frequency used. Therefore, it is likely that SWAMPS will have overall underestimated water fraction for vegetated areas. In addition, both SWAMPS and the static water datasets have biases, uncertainties and inconsistency in their representative season/periods. Ideally, all these factors should have been carefully accounted for when fusing the datasets, though it might not be practically achievable. The information of relative changes of SWAMPS water area seems more important to me than the absolute values when merging the datasets.

The reason for matching the $fw_{max}$ of SWAMPS with the static inventories is based on the assumption that the area of sparsely vegetated inundation captured by SWAMPS follows the same seasonal cycle as the densely vegetated area within a 0.25 deg grid cell. While it is conceivable that densely vegetated wetlands only get inundated during the wet season, we have no data source allowing this differentiated correction. As a result, our conservative approach preserving the seasonal amplitude of SWAMPS seems most reasonable. Moreover, we agree with the reviewer it would be ideal to improve the fusing approach by considering the biases and uncertainties in each of the mapping products, which require more ground-based monitoring for wetland areal dynamics. We have addressed these issues in the Discussion for future studies.

(b) The use of GSW to identify and mask out inland open water bodies seems over- simple to me. Assuming a lake with seasonal inundation changes, the pixels detected as water for less than 50% of the months were not classified as inland water (section 2.2.2), but they may be part of the lake over the wet season.

We agree with the reviewer that the correction using GSW is simplified, such that WAD2M includes some seasonal lake in its estimate of vegetated wetland inundation. However, given that the current version of WAD2M has already been distributed and used in other studies such as Global Carbon Project Methane (Saunois et al., 2020) and FLUXNET-CH4 upscaling (Delwiche et al., 2021), we feel it is important to document the methodology for the current version of WAD2M. We share the reviewer's concern about adequately excluding open water from this dataset, because the current version may still lead to some double-counting of methane fluxes from wetlands and open water. Hence, we will improve the resolve and remove the monthly GSW areas from WAD2M in the next version, planned for the next Global Methane Budget.

(c) What caused the overestimation of water fraction in dry areas such as central Australia (Fig.2a, 3a)? Did the uncertainty associated with drylands also affect wetland areas?

The overestimation of water fraction in dry areas is due to the limitation of SWAMPS, which is affected by the low surface emissivity due to the cretaceous outcrops of limestone deposits for semi-arid and arid regions. We have added descriptions to address this issue.

(d) Is it possible to examine data quality and accuracy for an area where ground/aircraft- based wetland mapping is available?

We agree with the reviewer that it's ideal to examine the data quality throughout with ground/aircraft-based wetland mapping. However, there are limited number of wetland mapping products and it is difficult to reconcile such vastly different wetland datasets. Thus, we are only able to evaluate the spatial distribution with independent regional datasets for West Siberian Lowland and Amazon and the temporal distribution with different large-scale water proxy datasets.

Minor comments: (a) For Fig. 9: please provide statistics on correlations between WAD2M times series and the others.

We have included the correlation between WAD2M and other sources in Fig. 9.

(b) For Fig. 9: did you miss GRACE time series in the upper-left figure?

Corrected

(c) Line 91-92: please revise the sentence, which is not accurate.

We have revised the sentence to "*L-band (~1 GHz) synthetic aperture radar (SAR) sensors are suitable for large scale wetland mapping because of their ability to penetrate clouds and detect flooding beneath most vegetation canopies (Melack et al., 2004)*".

(d) Line 101-102: please revise to improve clarity.

We have revised the sentence to "*1) higher-spatial resolution optical sensors can only detect open water in the absence of clouds and vegetation while SAR measurements can penetrate cloud and dense canopies but have limited temporal coverage;*"

(e) Line 104: did you mean "inundation under snow"? Any reference to support this?

We have revised the sentence to "*1) higher-spatial resolution optical sensors can only detect open water in the absence of clouds and vegetation while SAR measurements can penetrate cloud and dense canopies but have limited temporal coverage;*"

(f) Line 193-194: "The coastline..." Not sure about the meaning. Please revise or clarify.

We have revised the sentence to "*The coastal region is defined as areas along the coastline within 4 pixels (~1 km) and was then intersected with the ocean-labeled pixels from MOD44WA1 to separate the ocean from inland water.*"

---

## Author Comment (AC2) · 30 Mar 2021

Zhang et al. generate a novel dataset of global wetland dynamics using the remotely- sensed dataset, SWAMPS, as a starting point. They then address its shortcomings by bringing in regional datasets for known issues like saturated wetlands in the high latitudes (e.g. using NCSCD) or trouble seeing through canopy in tropics (e.g. using CIFOR). I generally find what they have done to be quite logical. I think the suggestion of an ensemble approach for future versions would be very useful and was glad to see it suggested. At present, the approach here relies quite heavily on the performance of SWAMPS/GLWD/NCSCD/CIFOR being the best products out there. That is unlikely to be correct for all locations so more regional specific datasets could likely improve future versions of WAD2M. My biggest complaint about the paper was the graphics and specifically the colour schemes. This sounds trivial but it did make interpreting the figures challenging. I hope the authors take my comments there seriously. As I think my comments can be relatively easily dealt with I am suggesting minor revisions.

We thank the reviewer for the interest in our study and for the constructive and insightful comments that help us to improve our manuscript. Please see our point-to-point responses below.

- line 51: also conversion to rice agriculture?

The sentence has been revised to include conversion to rice agriculture.

- l57: does Ramsar include rivers/lakes/ponds as wetlands? That is how I read that but am not sure if that was what was intended.

Yes, Ramsar sites include permanent river/lakes/ponds as per Ramsar's broad definition of wetland. We have revised the statement to improve clarity.

- l65: Can you provide an example of a dataset that omits them?

We've modified the sentence to "*Boreal taiga forests and tropical floodplains, which are considered CH4-emitting areas given their seasonally inundated states and significant $CH_4$ transport pathway via tree stem (Barba et al., 2019; Pangala et al., 2017), are underestimated by many wetland mapping products (Junk et al., 2013) due to the lack of record in inventories and difficulty in detecting dense forest canopies in satellite-based products.*".

- l.91 - detect flooding beneath most vegetation canopies - this begs the question which canopies can it then not detect the flooding underneath?

We have revised the statement to "detect flooding beneath woody vegetation or sufficiently dense canopies with thickness on the same order as the wavelength" to clarify the sentence.

- l.92 - maybe specify the frequency you are talking about, could be construed to mean satellite return frequency or something rather than frequency in the electomagnetic spectrum, esp as the next sentence talks about temporal coverage.

We have specified the satellite return frequencies in the sentence.

- l. 104 - Is there any issue with high-latitude regions due to the satellite orbits or the low radiation in the winter that should be mentioned here?

We have added the issue in the sentence as the reviewer suggested.

- l. 143 - the problem with this approach is that you then end up potentially increasing the wetland area since you are assuming that SWAMPS is wrong and the wetland product is correct. Given how error prone large-scale wetland mapping can be I think this is likely a dangerous assumption. I don't really disagree with the approach though, as there is not likely a better way to include wetlands that SWAMPS doesn't see. I would suggest that somewhere it is noted that this approach could inflate wetland areas as it would then include wetlands that may be erroneous in the wetland inventory.

Thank you for the suggestion. We want to be fully transparent about the underlying assumption that static maps are likely closer to the true maximum wetland extant than SWAMPS. We have added the statement in the Discussions about the limitation of our approach and the potential biases that can be introduced by the static inventories.

- l. 145 - Hugelius has published a new effort for high latitude peatlands, while I understand that this work might have been completed prior to the publication of that work, I wonder if there is much change between the NCSCD than the newer paper (Hugelius et al. 2020)? I see Hugelius is on the author list.

Thank you for suggesting the new peatlands maps from Hugelius et al., (2020). We agree with the reviewer that the latest progress on regional/global inventories that have higher accuracy should be integrated into the WAD2M processing. However, given that the current version of WAD2M has been distributed and used in other studies such as Global Carbon Project Methane (Saunois et al., 2020) and FLUXNET-CH4 upscaling (Delwiche et al., 2021), we feel it is important to document present this version of the methodology. We have added a version number to help track any modification to WAD2M. Moreover, we will strongly consider integrating more recent inventories including Hugelius et al., (2020) into the next version of WAD2M planned for the next Global Methane Budget.

- l 149 - I wonder if the dataset used for the temperate regions could be improved upon by using updated datasets for more specific regions, e.g. for Canada (Mahdianpari et al. 2020). Since the wetland inventory seems important (my comment for line 143), it seems reasonable to try to get the very best inventories. I am not sure if GLWD is that (although it was a good first attempt).

Thank you for the suggestion. We have included this national dataset in the Table 1 and will consider including this dataset in calibrating the next version of WAD2M. Indeed, national inventories are among the accurate maps but are challenging to compare to SWAMPS because of their wetland typology.

- l. 181 - what proportion of SWAMPS 3.2. observations were 'valid'? How many valid observations were needed to be able to produce the monthly values?

We have added one sentence to address this question: *"The sample sizes for the monthly mean range from 14-22 days globally with more measurements during the summer months and less measurements during winter months."*

- l. 183 - What does it mean that they were later 'refiltered'?

The statement has been revised to clarify "For SWAMPS v3.2, the coastal zone was first filtered out using a Landsat-based 90-m mask of permanent ocean waters defined by the G3WBM Global Water

Body Map dataset (Yamazaki et al., 2015) and then filtered using the MODIS MOD44W product to keep it consistent with the processing of GSW dataset."

- l. 186 - When I tried to use the GSW for a project I found it to have large issues with stripping of the data due to satellite passes and data quality control. Was this problem also encountered and if so how was the influence of the stripping minimized?

We calculated the long-term maximum of GSW for each of the 0.25 deg grid cell to minimize the stripping issue. When summarizing the 18 relevant years of record of GSW, the striping over cloudy or image-sparse region nearly complete disappears.

- Also with GSW, since it is based on Landsat does this mean it is also just the open waterbodies (i.e. no canopy cover)? I think you say this on line 196 but not earlier in describing what GSW provides so I want to be sure.

Yes, GSW is an open water product. We have now described its feature clearly in Line 88 in the Introduction to remind the readers.

- L 211: I think this is getting to my question about how it compares to the newer Hugelius product.

The major difference is that the newer Hugelius et al., (2020) product has extended the estimate of peatlands to 23 degree north based on Hugelius et al., 2013 and two other map products: the global WISE30sec dataset (Batjes et al., 2016) and the global SoilGrid250m dataset (Hengl et al., 2017).

- L. 230 to get an idea of how good the datasets agree for areas of overlap, was any attempt made to see how CIFOR and GLWD compare against each other or NCSCD vs. GLWD? If the agreement is poor then it doesn't allow a lot of confidence in the use of these datasets.

The comparison for Amazon and West Siberian Lowland in this study shows a large difference among multiple datasets. Given that GLWD is based on a range of different atlases and maps, some of which are more than 20 years old, we decided to use the more recent CIFOR for the calibration for pan-tropical (<30N) regions and NCSCD for regions northern high latitudes. GLWD is used over latitude bands that are not covered by CIFOR or NCSCD.

- Table 1 - misplaced comma final column and row.

Corrected

- It would be useful to explain why SWAMPS was the starting remotely-sensed dataset and not GIEMS. Why one over the other? Assumedly there was some performance improvement, like is seen in Fig 4a where SWAMPS seems more sensitive to high latitude wetlands?

Although there are notable differences between SWAMPS and GIEMS, the primary reason for choosing SWAMPS is because it was the only one updated to the end of the year 2018 when starting this project. Mapping wetland until 2018 was a requirement of ours in order to then simulate wetland methane flux over the same time-window as other components of the methane budget (Saunois et al. 2020).

- In Fig 1 the Inundation box dead-ends. So that information is not integrated into the final output? I think a line is missing. Also isn't the correcting factor applied to dynamic inundation, not becoming the corrected dynamic inundation as is shown? This figure is really hard to follow. I suggest a re-think.

We have modified the Figure 1 to clarify the logic chain.

- The maps have a, creative, colour scheme. Have you considered a linear colour map? There are good resource around like, https://sciviscolor.org/. At present there is an arbitrary strong visual cut-off at between 0.1 and 0.2 giving some sort of importance to a threshold that has none. Is the maximum value in all of the dataset 0.6? Also the maps could have Antartica cut off to allow a larger representation of the non-ice covered parts of the world. Also it would be nice if the images could have improved resolution. When I tried to zoom into see the distribution it looks as though the plotting used a setting that interpolated between gridcells, i.e. pixels are washed out. Please check and correct this. BTW, Fig 10 with the number of sources panel is the only time I think the colour scheme makes sense. Here it is categorical so the use of different colours with no continuous colour scheme makes perfect sense (small point on that one - why have a 0 value? Isn't that only for Antarctica which should be trimmed anyway?).

We have modified the plots throughout the manuscript with a linear color scheme.

- Fig 5, there is a hashtag you might want to check out #endrainbow. It is based on the fact that rainbow colour scheme tends to distort the perception of the data since values (e.g. yellows) with no greater importance stand out more to readers, but shouldn't as there is nothing important about the 0.5-0.6 fraction. Please consider using colour schemes that follow best practices for data visualization.

We have modified the plots with the linear color scheme.

- Fig 5 when the grey value is below 0 on the colour bar, how do we interpret that? It is missing/no values or 0? I assume it must be missing/no since Ternetieva isn't really showing wetlands then none for the most S and N part of the domain. But then other plots have the 0/missing/no wetland grey in the middle of their domains, so what is going on?

We have masked out the NA value in the plots to clarify the values in the maps.

- L 360 - 'owing to'

Corrected.

- Fig 6 - Sorry to continually complain about colour schemes but why not just use a linear blue colour scale? Why add in the green? It makes me think there is something especially important about middle of the road values, which isn't the case, so please reconsider. E.g. the most visually striking box in the lower plot (WSL) is for WAD2M and GSW - is that really the thing you want to stick in the reader's mind?

We have modified the color scheme with the brighter color representing higher Kappa value.

- Fig 8 over the same years? What years are they?

We have added descriptions in the caption.

- Fig 9 the caption doesn't mention if these are the watershed values and how that might be defined. Might be worth including a small sub-map showing the boundaries.

We have added text in the caption to describe the watershed maps used in the analysis.

- l388 - I think the comparison with Parker et al. 2018 is a bit tenuous, wetland extent is only part of what affects CH4 emissions so the consistency could just be fortuitous. Also this confuses me a bit, it says no significant trend for pan-tropical then the next sentence says tropical have net reduction. So how does pan-tropical differ from tropical?

We agree with the reviewer that the comparison is not robust and have removed the corresponding text.

- L430- this sentence seems to be saying two things: 1) further work needed to confirm and 2) improvement over existing. I suggest rewording so it is more one thought rather than a mash of two that conflict.

We have revised the text to "*Our study has demonstrated that the spatial distributions of WAD2M based on the data-fusion approach show reasonable improvements over existing wetland products and the temporal variations of WAD2M adequately represent interannual variation in response to the climate events such as El Niño-Southern Oscillation.*"

- L443 - I quite like this point about ensemble maps. I think it would make version 2.0 much better.

Thank you for the comments.

- L457 - artificial neural networks are a machine learning approach.

We have revised the text to "Downscaling can also be used to improve spatial resolution using machine learning approaches (Alemohammad et al., 2018; Kratzert et al., 2018; Wu et al., 2017) or physically-based hydrological models (Gumbricht, 2018)"

- L461 - what are less informed wetlands?

We have modified the text to "relatively under-examined wetlands"

- Some regions may be 'not wetland' then become wetland for, say a month since that is the shortest time this dataset can do, is the assumption that the system can behave like a wetland in that amount of time? Since the models will run their wetland model over this as a mask I think that is an underlying assumption but I am not sure if it is true. This is more of a comment than question and I am sure that number of pixels like that are minimal but I think the assumption is made that each time a pixel 'becomes' a wetland it is assumed to immediately adopt a wetland behaviour from a biogeochemical perspective. This is likely out of the scope of this paper but it does play into how this dataset is used.

That is correct. Some areas that are inundated for only a single month over 2000-2018 only are described as inundated wetlands for the purpose of this paper. From a biogeochemical standpoint these 1-month wetlands do not behave the same as a seasonally inundated wetland. That said, it would be entirely possible for biogeochemical modelers to only consider areas inundated for 2+ consecutive months, for example, as methane-emitting wetlands from WAD2M.

Refs cited:

Hugelius, G., Loisel, J., Chadburn, S., Jackson, R. B., Jones, M., MacDonald, G., Marushchak, M., Olefeldt, D., Packalen, M., Siewert, M. B., Treat, C., Turetsky, M., Voigt, C. and Yu, Z.: Large stocks of peatland carbon and nitrogen are vulnerable to permafrost thaw, Proc. Natl. Acad. Sci. U. S. A., 117(34), 20438–20446, 2020.

Mahdianpari, M., Salehi, B., Mohammadimanesh, F., Brisco, B., Homayouni, S., Gill, E., DeLancey, E. R. and Bourgeau-Chavez, L.: Big Data for a Big Country: The First Generation of Canadian Wetland Inventory Map at a Spatial Resolution of 10-m Using Sentinel-1 and Sentinel-2 Data on the Google Earth Engine Cloud Computing Platform, Can. J. Remote Sens., 1–19, 2020.

References:

Batjes, N. H.: Harmonized soil property values for broad-scale modelling (WISE30sec) with estimates of global soil carbon stocks, Geoderma, 269, 61–68, https://doi.org/10.1016/j.geoderma.2016.01.034, 2016.

Hengl, T., Mendes de Jesus, J., Heuvelink, G. B. M., Ruiperez Gonzalez, M., Kilibarda, M., Blagotić, A., Shangguan, W., Wright, M. N., Geng, X., Bauer-Marschallinger, B., Guevara, M. A., Vargas, R., MacMillan, R. A., Batjes, N. H., Leenaars, J. G. B., Ribeiro, E., Wheeler, I., Mantel, S., and Kempen, B.: SoilGrids250m: Global gridded soil information based on machine learning, PLOS ONE, 12, e0169748, https://doi.org/10.1371/journal.pone.0169748, 2017.

Delwiche, K. B., Knox, S. H., Malhotra, A., Fluet-Chouinard, E., McNicol, G., Feron, S., Ouyang, Z., Papale, D., Trotta, C., Canfora, E., Cheah, Y.-W., Christianson, D., Alberto, M. C. R., Alekseychik, P., Aurela, M., Baldocchi, D., Bansal, S., Billesbach, D. P., Bohrer, G., Bracho, R., Buchmann, N., Campbell, D. I., Celis, G., Chen, J., Chen, W., Chu, H., Dalmagro, H. J., Dengel, S., Desai, A. R., Detto, M., Dolman, H., Eichelmann, E., Euskirchen, E., Famulari, D., Friborg, T., Fuchs, K., Goeckede, M., Gogo, S., Gondwe, M. J., Goodrich, J. P., Gottschalk, P., Graham, S. L., Heimann, M., Helbig, M., Helfter, C., Hemes, K. S., Hirano, T., Hollinger, D., Hörtnagl, L., Iwata, H., Jacotot, A., Jansen, J., Jurasinski, G., Kang, M., Kasak, K., King, J., Klatt, J., Koebsch, F., Krauss, K. W., Lai, D. Y. F., Mammarella, I., Manca, G., Marchesini, L. B., Matthes, J. H., Maximon, T., Merbold, L., Mitra, B., Morin, T. H., Nemitz, E., Nilsson, M. B., Niu, S., Oechel, W. C., Oikawa, P. Y., Ono, K., Peichl, M., Peltola, O., Reba, M. L., Richardson, A. D., Riley, W., Runkle, B. R. K., Ryu, Y., Sachs, T., Sakabe, A., Sanchez, C. R., Schuur, E. A., Schäfer, K. V. R., Sonnentag, O., Sparks, J. P., Stuart-Haëntjens, E., Sturtevant, C., Sullivan, R. C., Szutu, D. J., Thom, J. E., Torn, M. S., Tuittila, E.-S., Turner, J., Ueyama, M., Valach, A. C., Vargas, R., et al.: FLUXNET-CH4: A global, multi-ecosystem dataset and analysis of methane seasonality from freshwater wetlands, 1–111, https://doi.org/10.5194/essd-2020-307, 2021.